# Kramers nodal lines in intercalated TaS$_2$ superconductors

Yichen Zhang [1,20], Yuxiang Gao [1,20], Aki Pulkkinen [2,20], Xingyao Guo[3], Jianwei Huang [1], Yucheng Guo [1], Ziqin Yue [1,4], Ji Seop Oh[1,5], Alex Moon[6,7], Mohamed Oudah [8], Xue-Jian Gao [3], Alberto Marmodoro [2], Alexei Fedorov [9], Sung-Kwan Mo [9], Makoto Hashimoto [10], Donghui Lu [10], Anil Rajapitamahuni [11], Elio Vescovo [11], Junichiro Kono [1,12,13,14,15], Alannah M. Hallas [8,16,17], Robert J. Birgeneau [5,18], Luis Balicas [6,7], Ján Minár [2], Pavan Hosur[19], Kam Tuen Law [3], Emilia Morosan [1,12,15] ✉ & Ming Yi [1,12,15] ✉

Kramers degeneracy is one fundamental embodiment of the quantum mechanical nature of particles with half-integer spin under time reversal symmetry. Under the chiral and noncentrosymmetric achiral crystalline symmetries, Kramers degeneracy emerges respectively as topological quasiparticles of Weyl fermions and Kramers nodal lines (KNLs), anchoring the Berry phase-related physics of electrons. However, an experimental demonstration for ideal KNLs well isolated at the Fermi level is lacking. Here, we establish a class of noncentrosymmetric achiral intercalated transition metal dichalcogenide superconductors with large Ising-type spin-orbit coupling, represented by In$_x$TaS$_2$, to host an ideal KNL phase. We provide evidence from angle-resolved photoemission spectroscopy with spin resolution, angle-dependent quantum oscillation measurements, and ab-initio calculations. Our work not only provides a realistic platform for realizing and tuning KNLs in layered materials, but also paves the way for exploring the interplay between KNLs and superconductivity, as well as applications pertaining to spintronics, valleytronics, and nonlinear transport.

Symmetry plays a ubiquitous role in dictating the electronic properties of solids, enriched by the introduction of topology into the field of condensed matter[1]. In particular, recent developments have recognized the presence of topological degeneracies in the electronic band structure originating from crystalline symmetries, such as non-symmorphic[2,3], chiral[4,5], and achiral[6] operations, which could be further intertwined with magnetic[7–9] and charge order[10]. In the scenario of non-symmorphic symmetry, Dirac- and Weyl-type band crossings[11], unconventional multi-fold fermions[12–14], and hourglass fermions[15–17] could arise from the glide-mirror or screw-axis symmetries. Meanwhile, chiral and noncentrosymmetric achiral little group symmetries emphasize the absence of inversion, albeit bearing overlapping space groups with the non-symmorphic ones. Crucially, mirror or roto-inversion symmetries must be present (absent) in the achiral (chiral) structure. This difference is the key in determining the distinction between Kramers-Weyl fermions pinned at time-reversal invariant momenta (TRIM) in chiral crystals[4] and the type-I Kramers nodal lines (KNLs) connecting TRIM in noncentrosymmetric achiral crystals[6]. These symmetries, if bearing topological quasiparticles located near the Fermi level and isolated from trivial bands, could generate distinct transport, thermal, and optical phenomena such as Berry curvature-related anomalous Hall[18] and Nernst effects[19], chiral anomaly[20], dissipationless edge current[21], and circular photogalvanic effect[22]. Recently, chiral crystals have garnered renewed interest owing to their

monopole-like orbital angular momentum texture[23], leading to promising aspects on orbital magnetotransport, while noncentrosymmetric achiral crystals hosting KNLs, a type of Dirac solenoid concentrating quantized Berry flux of $\pi$, still require identification of ideal material candidates and unequivocal experimental demonstration.

Since the theoretical prediction of KNLs[6], experimental work has suggested a few material platforms. These include time reversal symmetry breaking superconductors of the $T$RuSi ($T$ = Ti, Ta, Nb, and Hf)[24] and the LaPtSi[25] family, the paramagnetic state of SmAlSi[26,27], and charge density wave (CDW)-driven KNLs in rare-earth tritellurides[28,29]. However, in the case of rare-earth tritellurides, the approximated polar supercell derived from the x-ray diffraction (XRD) resolved incommensurate CDW superspace group[28,30] directly contradicts the observation of preserved inversion symmetry in the combined studies of

Raman spectroscopy, rotational-anisotropy second harmonic generation, and other experimental techniques[31], hence casting doubt on the existence of KNLs driven by the CDWs in the series of systems. Moreover, these materials are all either multi-band systems near the Fermi level, $E_F$, or have the KNLs located far away from $E_F$.

Here, through comprehensive physical and thermodynamic properties characterization, angle-resolved photoemission spectroscopy (ARPES), and spin-resolved ARPES measurements, ab-initio calculations, and quantum oscillation measurements, we introduce a material class that exhibits an ideal KNL metallic phase, in the form of a noncentrosymmetric achiral intercalated transition metal dichalcogenide (TMD) family in the space group of $P\bar{6}m2$[32,33]. These are exemplified by $In_xTaS_2$ ($x = 1/2$ and 1) (Fig. 1), both of which also feature superconducting ground states[34–36]. In addition, similar physics of the KNLs in the isostructural intercalated TMD superconductors, $In_{2/3}TaS_2$

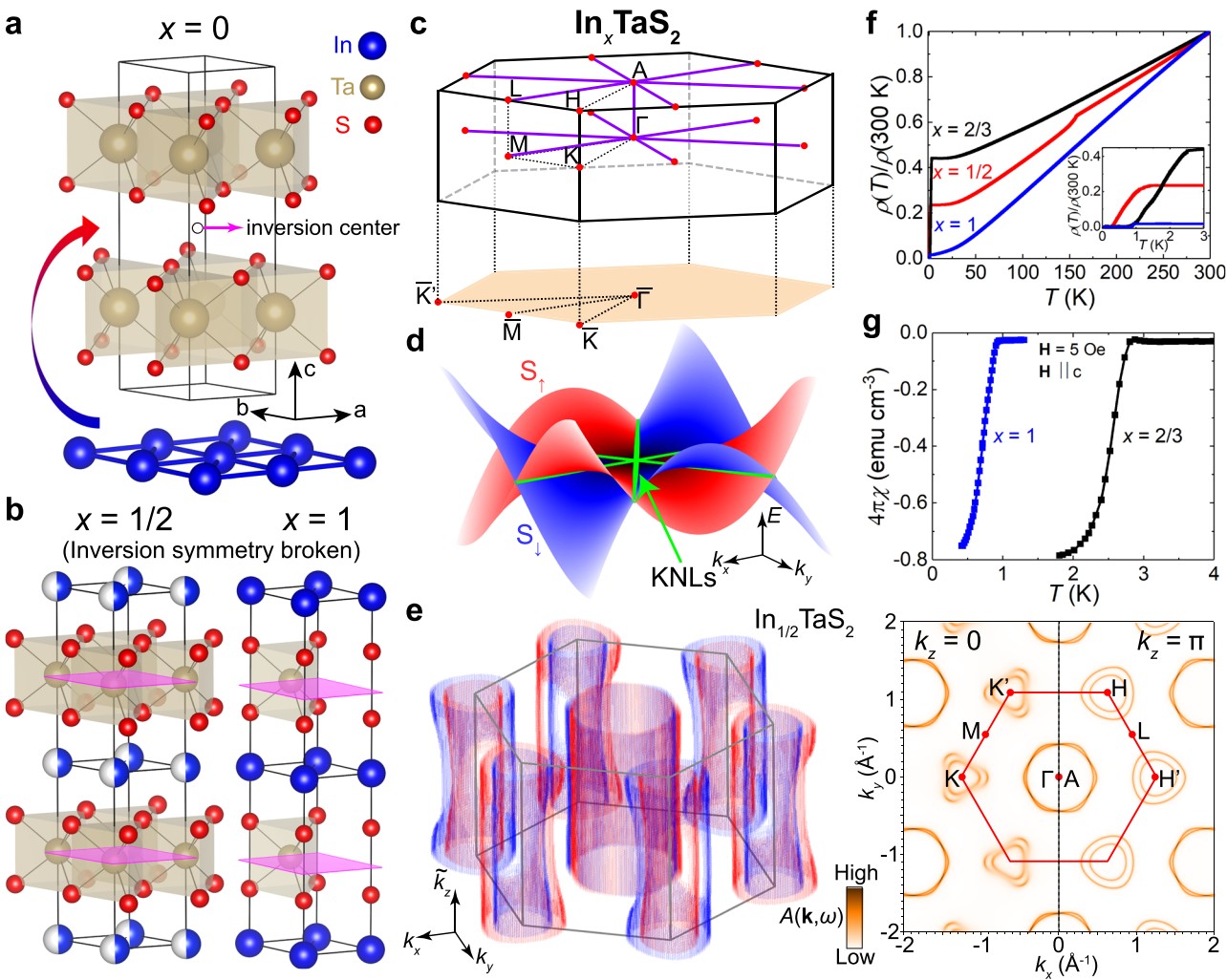

**Fig. 1 | Design of ideal Kramers nodal lines in intercalated transition metal dichalcogenides and their material properties. a** Illustration of the indium intercalation into the inversion symmetric 2$H$-TaS$_2$ (with one inversion center denoted by the empty circle). **b** Crystal structures of In$_x$TaS$_2$ (left: $x = 1/2$, right: $x = 1$). Both exhibit broken inversion symmetry. The transparent magenta horizontal planes denote the $M_z$ mirror symmetry that is crucial to the formation of the Kramers nodal lines (KNLs). **c** Three-dimensional (3D) and two-dimensional (2D) projected surface Brillouin zone of the crystal structure in (**b**), where the purple lines indicate the KNL directions ($M$–$\Gamma$–$A$–$L$). **d** Low-energy dispersions of the spin-orbit coupling splitting (red and blue) and Kramers nodal lines (green tubes) of the point group of the intercalated transition metal dichalcogenide family, D$_{3h}$ (G$^{11}_{24}$: R$_7$, R$_8$)[6]. $S_\uparrow$ and $S_\downarrow$ represent pseudospin up and down. The Hamiltonian takes the form

of $i\alpha_1(k_+^2 - k_-^2)k_z\sigma_x - \alpha_1(k_+^2 + k_-^2)k_z\sigma_y + i\alpha_2(k_+^3 - k_-^3)\sigma_z$, where $\alpha_1 = -0.5$, $\alpha_2 = 0.2$, $k_z = 0$, and $k_\pm = k_x \pm ik_y$. **e** Ab initio 3D voxel-style Fermi surface of In$_{1/2}$TaS$_2$ (left) and 2D Bloch spectral function (BSF) calculated at the $k_z = 0$ and $\pi$ slices at the Fermi level (right). The 3D red and blue voxels denote high-intensity values of the BSF, which has turned into a Lorentzian-like continuum due to the random occupation of the indium lattice site marked in blue/white in panel (**b**). The $\tilde{k}_z$ indicates that the $k_z$ direction is elongated for visualization. **f** Resistivity as a function of temperature for In$_x$TaS$_2$ (blue: $x = 1$, black: $x = 2/3$, red: $x = 1/2$). The inset is a zoom-in view of the low temperature data, showing the superconducting transitions for all three compounds. **g** The susceptibility as a function of In$_x$TaS$_2$ (blue: $x = 1$, black: $x = 2/3$). $4\pi\chi$ approaches the value of −1 at the lowest temperatures, indicating bulk superconductivity in both compounds.

and PbTaSe$_2$, is discussed in the Supplementary Information (notes 2 and 6). Due to the environment of the inversion symmetry broken TMD layers[37] (Fig. 1a, b), the KNL band topology is also coupled with spin-valley degrees of freedom (Fig. 1c, d). Such spin texture of KNLs in the intercalated TMD system under the D$_{3h}$ point group can be clearly visualized by the low-energy band structure only considering spin-orbit coupling (SOC) terms (Fig. 1d), of which the momentum-dependent relativistic pseudospin splitting pattern is reminiscent of the recently generalized concept of nonrelativistic altermagnetism[38,39] but achieved without time reversal symmetry breaking. Meanwhile, the fact that the directional relativistic splitting is locked with the crystal structure symmetry, unlike the nonrelativistic altermagnetic splitting associated with the ligand-environment-enriched anti-ferromagnetism, enables experimental investigation without complications of domain alignment. Furthermore, our theoretical analysis on the generic ideal KNL model at the Fermi level reveals several potential magnetic field-induced effects, including anomalous Hall effect (AHE) and chiral Majorana modes that are theoretically possible in the $s$-wave superconducting phase. Additionally, the noncentrosymmetric achiral $P\bar{6}m2$ In$_x$TaS$_2$ associates the symmetries of the KNL material family with recent experimental reports of giant anomalous nonlinear transport[40] and strain-induced superconducting diode effect[41] in the isostructural PbTaSe$_2$.

## Results

### Design of Kramers nodal line metals and characterization of the In$_x$TaS$_2$ family

The first step in constructing a noncentrosymmetric achiral crystal structure is to break the inversion symmetry. 2$H$-TaS$_2$ and other TMDs of the 2$H$ phase have been long known to have a centrosymmetric structure[42] (see Fig. 1(a) for one exemplary inversion center). However, due to the local broken inversion symmetry constrained within a single layer, layer-resolved hidden spin polarization has been reported by surface-sensitive ARPES techniques[43,44]. In addition, extensive efforts focused on the monolayer and few-layer regime have shown evidence for unconventional Ising superconductivity that allows both spin-singlet and spin-triplet pairings[45–51]. To achieve inversion symmetry breaking in the bulk, we have intercalated 2$H$-TaS$_2$ with trigonal indium layers, as shown in Fig. 1a, b. The indium intercalation changes the 2$H$ stacking to 1$H$ stacking, and consequently, leads to broken inversion symmetry in the bulk of the In$_x$TaS$_2$ crystal structure. We also note that, as illustrated in Fig. 1b, there is a structural difference between $x =$ 1/2, 2/3 and $x = 1$: the In atoms are located above the Ta atoms for $x =$ 1/2 and 2/3 (left, Fig. 1b), while for $x = 1$, the In atoms are above the S atoms (right, Fig. 1b). Such structural difference is supported by Rietveld refinements on the powder crystal XRD (see Supplementary Note 1). As a result, the Weyl nodal rings predicted in InTaS$_2$ near $H$ point[35] (see Fig. 1c for the Brillouin zone (BZ) definition) based on a structure of In on top of Ta would instead become predicted Weyl points near $H$[36]. Nonetheless, the KNL from the noncentrosymmetric achiral little group symmetry in In$_x$TaS$_2$ is fundamentally different from all the topological crossings predicted previously for similar compounds[35,36,52,53]. In Fig. 1b, we highlight the most important symmetry operation for the formation of the KNLs in the In$_x$TaS$_2$ family as magenta planes, the $M_z$ mirror. Under the point group symmetry of In$_x$TaS$_2$, the electronic band structure from a minimal two-band low-energy Hamiltonian at $k_z = 0$[6] only considering SOC terms shows an alternating pattern of pseudospin splitting, as shown in Fig. 1d, while the green tubes represent the topologically nontrivial KNLs. The complete distribution of the KNLs is indicated by the purple lines in the first Brillouin zone (BZ) in Fig. 1c. Furthermore, we visualize the realistic Fermi surface (FS) Bloch spectral function (BSF) of In$_{1/2}$TaS$_2$ calculated from first-principles at $k_z = 0$ and $k_z = \pi$ in Fig. 1e (right) showing the Kramers degeneracy along $\Gamma$−$M$ and $A$−$L$ whenever the two hexagonal-like FS sheets centered around $k_x = k_y = 0$ intersect. The

three-dimensional (3D) FSs extracted from high intensities of the BSF cast in voxel-style plot in Fig. 1e (left) reflect the quasi-two-dimensional (2D) nature of In$_{1/2}$TaS$_2$ at the Fermi level.

In addition to the broken inversion symmetry and the KNLs, indium intercalation also changes the superconducting properties of the TaS$_2$ layers. The bulk 2$H$-TaS$_2$ becomes superconducting below 0.5 K[46]. As shown in Fig. 1f, the resistivity curves of In$_x$TaS$_2$ ($x =$ 1/2, 2/3, 1) indicate a superconducting ground state, with $T_C$ (defined as $R(T_C) =$ 0.9$R_0$) varying with $x$ from 1.1 K for $x =$ 1/2, 2.3 K for $x =$ 2/3, to 0.9 K for $x = 1$. In addition, the bulk superconductivity of In$_x$TaS$_2$ ($x =$ 2/3, 1) is confirmed by magnetization measurements, as shown in Fig. 1g. Insights into the $T_C$ enhancement of In$_{2/3}$TaS$_2$ when compared to those of In$_{1/2}$TaS$_2$ and InTaS$_2$ can be gained from a calculation of the momentum-resolved BSF and site-resolved density of states, which shows an increased proximity of In electronic states to $E_F$ for $x =$ 2/3 (see discussion in Supplementary Note 2). The resistivity curve of In$_{1/2}$TaS$_2$ also reveals a transition around 150 K in Fig. 1f, previously suggested to be a charge density wave (CDW)-like transition[35], while absent in In$_{2/3}$TaS$_2$ and InTaS$_2$. However, as we elaborate in the following, the measured high-quality ARPES band dispersions of In$_{1/2}$TaS$_2$ at 15 K do not resolve any evidence for band folding or gap features within the experimental resolution that affect the nodal line structure. This implies negligible effects of the potential CDW transition on the electronic band structure, which, therefore, does not modify the KNL symmetry requirements and observables, consistent with previous bulk structural symmetry characterizations on a refined occupation of In$_{0.49}$TaS$_2$ down to 12 K[33].

### Ideal Kramers nodal line metal In$_{1/2}$TaS$_2$ with spin-valley polarization

To demonstrate the physics of KNL and its spin-orbital texture, a candidate material of ideal KNL metals is highly desired. Here, "ideal KNL" is defined as an isolated KNL band that crosses $E_F$, with a large SOC that results in a band splitting away from the KNL momentum directions. Among the noncentrosymmetric achiral intercalated TMD material family, we identify In$_{1/2}$TaS$_2$ to exhibit such ideal properties. First, to demonstrate the KNL properties, we carried out ab-initio calculations on the bulk BSF of In$_{1/2}$TaS$_2$, where the coherent potential approximation was used on the In sites accounting for the random vacancy without extending to a CDW supercell. As outlined by the red box in Fig. 2a, the calculation shows a single set of isolated bulk KNL bands along $M$−$\Gamma$−$A$−$L$ crossing the Fermi level. Away from this momentum path, the KNL band splits into two spin-orbital branches, as shown along $\Gamma$−$K$−$M$ and $A$−$H$−$L$, where the $k_z = \pi$ dispersions show the most prominent splitting. Projecting Green's function with the $\sigma_z$ operator in Fig. 2b, it is unambiguously shown that the spin is strictly degenerate along the KNLs and highly polarized in $s_z$ along the off-KNL directions due to the Ising SOC. The splitting is most clearly demonstrated along $A$−$H$−$L$, but hybridized with In orbitals along $K$−$M$. To directly observe the ideal KNL band topology and provide the smoking-gun evidence for the spin splitting, we performed spin-integrated ARPES (Fig. 2) and spin-resolved ARPES (Fig. 3) measurements on In$_{1/2}$TaS$_2$, of which the same sample was exfoliated and examined by scanning electron microscopy with energy dispersive x-ray spectroscopy to confirm spatial homogeneity and the In concentration (see Supplementary Note 3). As shown in Fig. 2c, the In$_{1/2}$TaS$_2$ FSs consist of two concentric pockets around $\bar{K}$ and $\bar{K}'$ and two hexagonal pockets centered around $\bar{\Gamma}$, with no clear evidence of CDW folding or gaps. The ARPES band dispersions along $\bar{\Gamma} - \bar{M} - \bar{K} - \bar{\Gamma}$ presented in Fig. 2f directly demonstrate the doubly-degenerate KNL crossing at $E_F$ along $\bar{\Gamma} - \bar{M}$ and its clear splitting along $\bar{M} - \bar{K} - \bar{\Gamma}$. The chosen 75 eV of photons here is justified to be close to the $k_z = 0$ plane through a photon-energy-dependent scan focusing on certain $k_z$ sensitive features at deeper binding energies, as presented in Supplementary Fig. 4 of the Supplementary Note 4. To theoretically capture

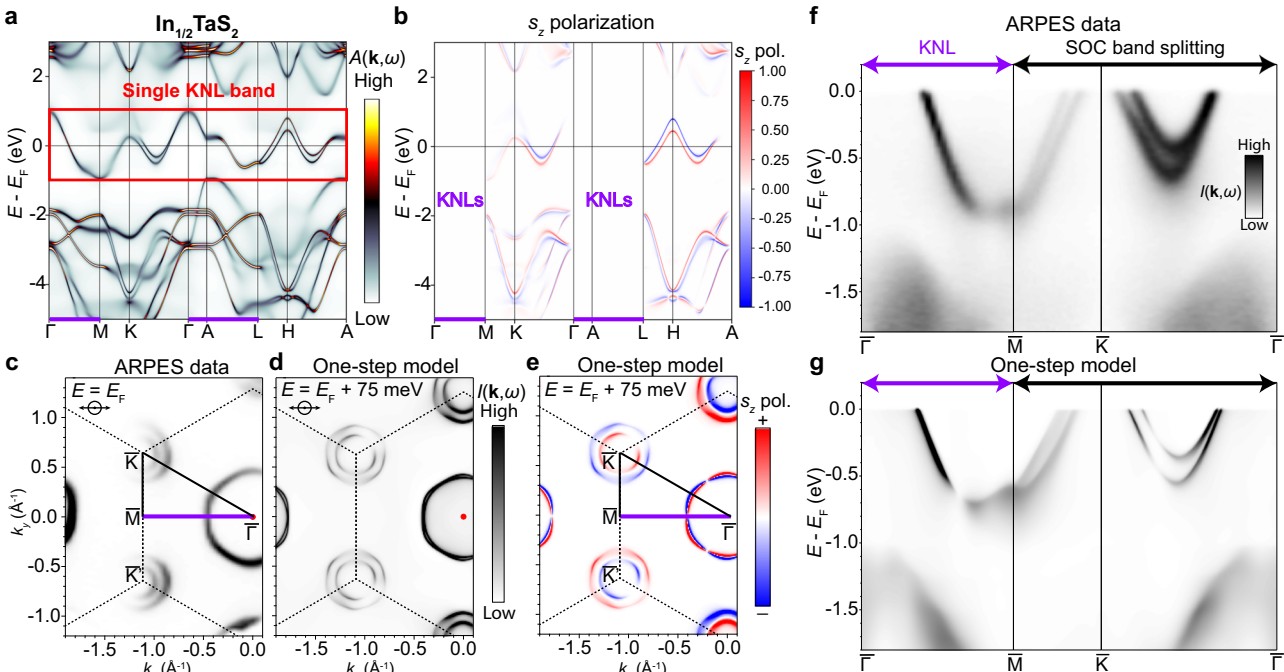

**Fig. 2 | Demonstration of an ideal Kramers nodal line (KNL) metal, In$_{1/2}$TaS$_2$.**
**a** Bulk Bloch spectral function of In$_{1/2}$TaS$_2$ calculated from the full-potential fully-relativistic Korringa–Kohn–Rostoker (KKR) method, highlighting the single KNL band crossing the Fermi level. **b** Spin projection along $z$ for the bulk band dispersion in (**a**). **c** Fermi surface of In$_{1/2}$TaS$_2$ measured by angle-resolved photoemission spectroscopy (ARPES). The KNL direction $\bar{\Gamma} - \bar{M}$ is denoted by the purple line. All experimental data in the figure were taken with 75 eV photons of $p$ polarization as indicated at 15 K. **d** One-step calculation of the Fermi surface using the same experimental conditions, where the theoretical Fermi level has been shifted up by 75 meV to achieve the best match with the data. **e** Spin polarization along $z$ extracted from the same one-step calculation in (**d**). **f**, **g** Electronic band structure along $\bar{\Gamma} - \bar{M} - \bar{K} - \bar{\Gamma}$ showing the KNL dispersions measured in ARPES and predicted by one-step calculations. $s_z$ pol.: $s_z$ polarization. SOC: spin-orbit coupling.

the observed ARPES spectra with high accuracy, state-of-the-art one-step photoemission calculations for the FS and band dispersions along $\bar{\Gamma} - \bar{M} - \bar{K} - \bar{\Gamma}$ are carried out with the In termination (see Fig. 2d, g), showing an excellent agreement with ARPES data, confirming the validity of the random vacancy modeling on the In sites. An in-depth discussion for the termination-dependent band structure of In$_{1/2}$TaS$_2$ is presented in Supplementary Fig. 7 of the Supplementary Note 5, supporting the adopted In termination. Furthermore, the spin-projected one-step calculation indicates a spin-polarized FS encoding spin-valley degree of freedom at $\bar{K}$ and $\bar{K}'$ (Fig. 2e).

In the In$_{1/2}$TaS$_2$ family, the spin-valley polarization arises from the inversion symmetry breaking[37] which is satisfied as a prerequisite of the KNL little group symmetries. Meanwhile, the mirror symmetries such as the magenta ones denoted in Fig. 1b in the noncentrosymmetric achiral little group generate KNLs concentrating Berry curvature and forcing spin degeneracy robust against SOC. Further, heavy elements such as Ta provide strong strength of the SOC, exhibiting an Ising-type splitting at valleys such as $K$ and $K'$ mainly due to the broken $\Gamma - K - H - A$ mirror[54]. Experimentally, the investigation of the spin-valley locking behavior associated with the KNLs using spin-resolved ARPES is presented in Fig. 3. As the spin polarization of the split bands should reverse in the two opposite directions away from the KNL, we can directly measure the spin polarization across a KNL. We carried out spin-resolved measurement of two pairs of momentum points on opposite sides of the $\Gamma - M$ KNL. $k_1$ and $k_2$ are along $\bar{K} - \bar{M} - \bar{K}'$ (Fig. 3a). The spin up and down energy distribution curves (EDCs) selectively show the spin texture along the out-of-plane direction ($s_z$), and we plot them in direct comparison to the EDCs measured in spin-integrated mode in Fig. 3b. Even though for both $k_1$ and $k_2$, the band closer to $E_F$ exhibits weaker intensity as shown in the spin-integrated EDCs, the spin-resolved EDCs still show clear distinction for the split bands, namely for $k_1$ the lower binding energy band is dominantly spin up

while the higher binding energy band is dominantly spin down. Furthermore, we can plot the $s_z$ polarization defined as $P_z = \frac{1}{S}\frac{I_\uparrow - I_\downarrow}{I_\uparrow + I_\downarrow}$ (Fig. 3c, $S$ being the Sherman function, $I_\uparrow$ and $I_\downarrow$ the spectral intensity measured in the spin up and spin down channels, respectively), which clearly shows the opposite spin polarization for the two bands split from the KNL. $k_2$, on the opposite side of the $\Gamma - M$ KNL, shows reversed spin polarization of the two split bands (Fig. 3d, e). To demonstrate this spin polarization reversal even more clearly, we show similar measurements for a pair of momentum points slightly away from the $\bar{K} - \bar{M} - \bar{K}'$ high symmetry direction where the photoemission matrix elements allow comparable intensity of the two split bands as observed on each spin-integrated EDC ($k_3$ and $k_4$) (Fig. 3f–j). The spin-resolved ARPES data ($s_z$) measured here show an even better-defined peak structure and showcase consistent spin texture with spin polarization reaching as high as nearly 80%. Therefore, the combined experimental and theoretical results definitively demonstrate the coupled spin-valley polarization in In$_{1/2}$TaS$_2$, associated with the KNL band topology and the underlying Ising-type SOC shown previously only for monolayer TMD materials, which could potentially be relevant to applications in spintronics and valleytronics.

## Kramers nodal lines in the 3D electronic band structure of InTaS$_2$

As the noncentrosymmetric achiral symmetry is common across the family of intercalated TaS$_2$, the KNL should be a universal property of all members of this family. In this section, we present the evidence for KNLs for the stoichiometric InTaS$_2$ in the $P\bar{6}m2$ space group. In addition, we also present the ARPES experimental measurement of the KNL band structure of the isostructural PbTaSe$_2$ in the Supplementary Note 6. With full In intercalation, InTaS$_2$ is more three-dimensional and exhibits stronger $k_z$ band dispersions when compared to In$_{1/2}$TaS$_2$ (Fig. 1e). Due to the $k_z$ broadening effect of the photoemission process,

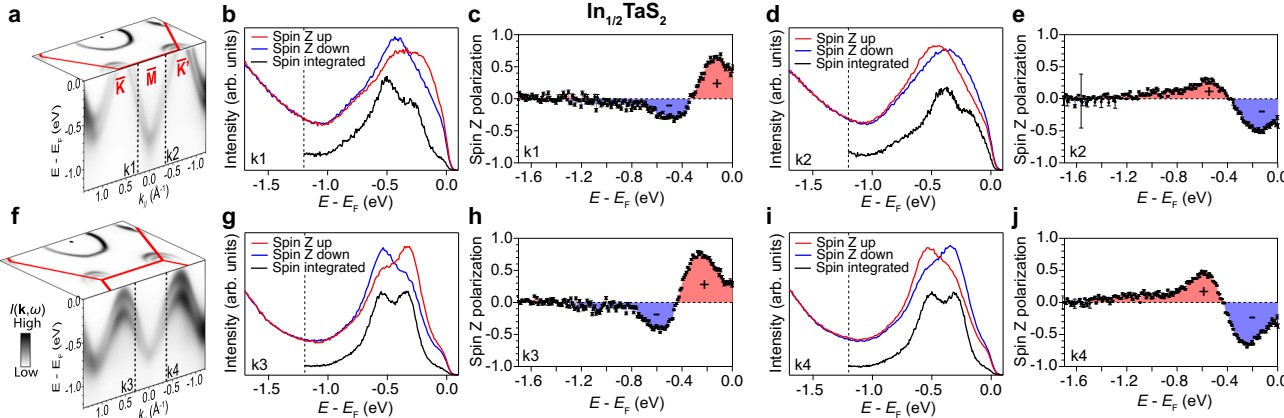

**Fig. 3 | Spin-resolved angle-resolved photoemission spectroscopy on In$_{1/2}$TaS$_2$.** **a** Three-dimensional view of the band dispersions ($k_x$-$k_y$ and $E$-$k_{//}$) indicating the momenta where the spin-resolved energy distribution curves (EDCs) were taken (vertical dashed lines in the $E$-$k_{//}$ plot) near the high symmetry direction $\bar{K} - \bar{M} - \bar{K}'$. **b**, **d** Spin-resolved EDCs taken at $k_1$ and $k_2$ on both sides of the $\Gamma-M$ Kramers nodal lines displaying reversed spin-polarized peak positions. Raw spin-integrated EDCs taken in the normal mode are attached as black curves at the bottom. **c**, **e** Converted spin polarization from (**b**) and (**d**), respectively. **f–j** Same as (**a**)-(**e**), but at $k_3$ and $k_4$ slightly away from the Brillouin zone boundary $\bar{K} - \bar{M} - \bar{K}'$. Error bars of the spin polarization are explained in the Methods. See also Methods for clarification of (**f**).

the measured FS of InTaS$_2$ (Fig. 4a) is more complex than the one of In$_{1/2}$TaS$_2$. Meanwhile, stronger $k_z$ integration in the photoemission process of InTaS$_2$ gives out weaker photon-energy-dependence for the band dispersions, as evidenced by the $h\nu$-scan along $\bar{\Gamma} - \bar{M}$ shown in Supplementary Fig. 5. To account for potential surface states and the strong $k_z$ broadening effect in the vacuum ultraviolet ARPES spectra, we undertook ab-initio calculations projected to both In and S surface terminations. We find that the measured electronic structure in Fig. 4a can be best described by the summation of the calculations for both terminations. This is likely due to the fact that the In and S terminations have equal probabilities during the cleaving process. Therefore, the ARPES beam spot would probe the superposition of band structures under both terminations, while the separate contributions from each termination are elaborated in Supplementary Fig. 6 of Supplementary Note 5. The direct comparison of the measured and calculated FSs is shown in Fig. 4a, b, where features such as the trefoil-like pattern centered at $\bar{K}$, the elliptical pockets centered at $\bar{M}$, and the $k_z$ broadened hexagonal pockets centered at $\bar{\Gamma}$ can be well reproduced. Importantly, we note that pairs of FSs are enforced to be degenerate whenever intersecting the $\Gamma-M$ direction, reflecting the properties of the KNL band topology. This can be further confirmed by the measured band dispersions along $\bar{\Gamma} - \bar{M} - \bar{K} - \bar{\Gamma}$ (Fig. 4c), where we also overlay the bulk band calculations for $k_z = 0$ and $\pi$. Clearly, both the bulk calculations and the ARPES data demonstrate that the bands along $\Gamma-M$ are doubly degenerate KNLs and split into two spin-orbital branches along $M-K-\Gamma$. To better capture the band dispersions observed in Fig. 4c beyond bulk contributions, we also display the surface calculations projected onto the In surface in Fig. 4d, which show an excellent match with the ARPES data.

Since the FS of InTaS$_2$ contains more 3D dispersions, as delineated in Fig. 5a, and ARPES has significant spectral intensity contributions from $k_z$ broadening and surfaces, we use quantum oscillation (QO) measurements to provide a more complete investigation of the 3D bulk FS topology. The assignment of the Fermi pockets discussed in QO measurements and density functional theory (DFT) calculations is illustrated in Fig. 5b for a cross section of the FS at $k_z = 0$. Next in Fig. 5c–h, the resistivity is measured as a function of the external magnetic field at different temperatures and field orientations, revealing Shubnikov de-Haas (SdH) oscillations. Strong SdH oscillations can be observed up to 20 K and 14 T after a smooth background subtraction, as can be seen in Fig. 5c. The subsequent fast Fourier transform (FFT) on the SdH oscillations reveals that they consist of two

frequencies F$_\alpha$ = 104 T and F$_\beta$ = 212 T (Fig. 5d). Despite the fact that F$_\beta$ is close to 2 × F$_\alpha$, the effective mass fitting (inset of Fig. 5d) shows that the two frequencies originate from distinct Fermi surface cross-sectional areas, since the effective mass m$_\beta$ is not twice that of m$_\alpha$. The small effective masses also imply that the underlying band dispersion is linear or close to linear.

To build a correspondence between QOs and the FS, we further measured SdH oscillations for InTaS$_2$ at different field orientations (Fig. 5e, f). The measurement configuration is shown in the inset of Fig. 5f. The SdH oscillations become significantly weaker when the field orientation is moved away from the $c$ axis (Fig. 5e), consistent with the expectation for van der Waals materials. We further compare the frequencies from the SdH oscillations (symbols) to the expected values from DFT calculations (dashed line) through the Onsager relationship: $F = \frac{\hbar A_{ext}}{2\pi e}$, where $\hbar$ is the reduced Planck constant, $A_{ext}$ is the extremal cross section of a Fermi pocket, and $e$ is the electron charge. A great match between theory and experiment can be established (Fig. 5f). Furthermore, F$_\beta$ is related to the Fermi pocket from the KNLs along $\Gamma-M$, as indicated by the arrow in Fig. 5b.

The complex Fermi surface of InTaS$_2$, as illustrated in Fig. 5a, b and 4a–d, should lead to QOs of various frequencies, while the SdH oscillations up to 14 T in Fig. 5c–f only consist of two frequencies. To further incorporate larger Fermi pockets, we measured the SdH oscillations up to 44.8 T at different magnetic field orientations (Fig. 5g, h, Supplementary Information Figs. 10, 11). The SdH oscillations (Supplementary Information Fig. 10a, b) show a much more complex spectrum with more frequencies, consistent with the complex Fermi surface of InTaS$_2$. Most interestingly, we discovered QOs of frequencies 6 kT and 12 kT (Fig. 5g). By comparing the experimental results (symbols) to DFT calculations (dashed lines) in Fig. 5h, we found that these frequencies are related to the oscillations from the Fermi pocket $\epsilon$ (see Fig. 5b) and its second harmonic. Such $\epsilon$ pockets encode pinched points enforced by both the $\Gamma-M$ and $\Gamma-A$ KNLs. Moreover, the frequency of the Fermi pocket $\epsilon$ (symbols) is significantly lower than the expected value for a cylindrical Fermi pocket (solid curve in Fig. 5h), which indicates that it must be a closed pocket as in the DFT calculation. We notice that at $\theta \sim 9$-$10°$, the fundamental oscillations from the Fermi pocket $\epsilon$ vanish while the second harmonic oscillations persist. This implies that at this field orientation, the quantum oscillations from the Fermi pocket $\epsilon$ are close to the spin-zero effect[55]. Overall, through ARPES and quantum oscillation measurements, we directly observe the Fermi pockets related to the KNLs in

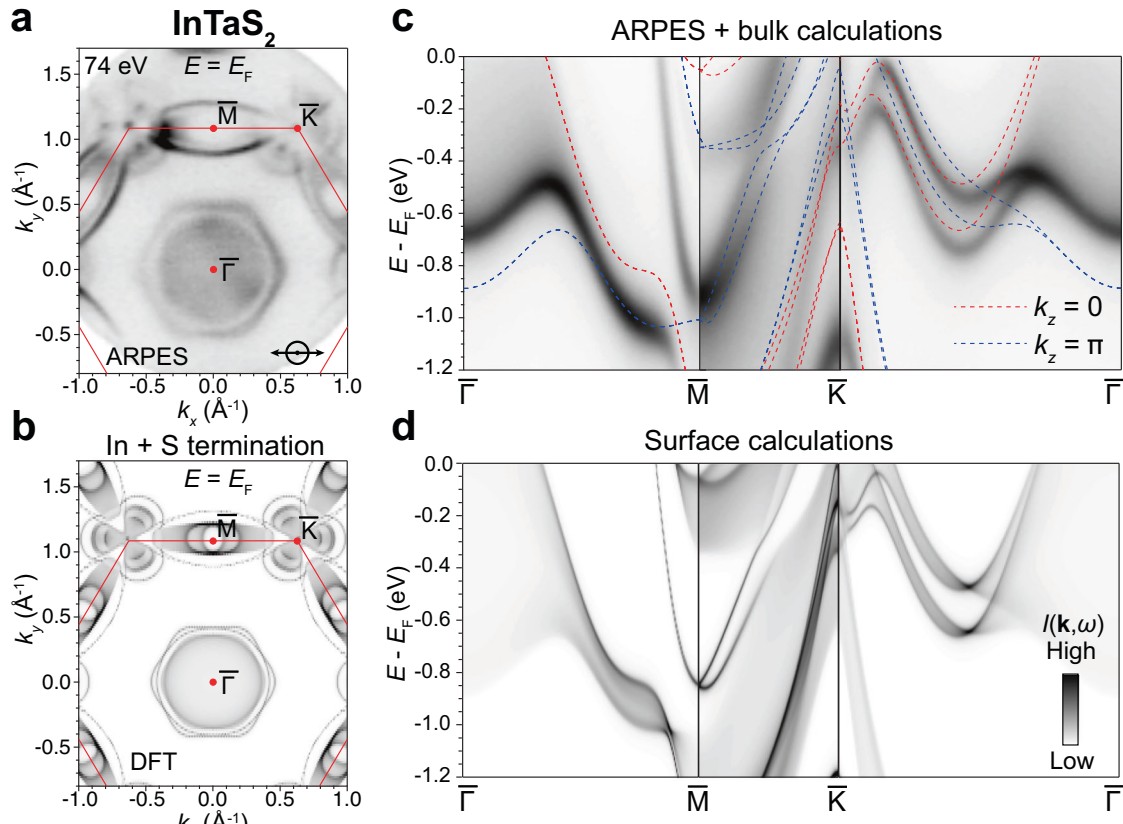

**Fig. 4 | Kramers nodal lines and electronic band structure in InTaS₂. a** Fermi surface (FS) of InTaS₂ measured by 74 eV *p* polarized photons (indicated at the bottom right of the panel) at 23.5 K. **b** FS calculated from first-principles using a superposition of both In and S terminations, where a surface onsite energy correction of −0.7 eV is applied to the In *s* orbitals and a +0.2 eV onto the Ta *d* orbitals. **c** Band dispersions along $\bar{\Gamma} - \bar{M} - \bar{K} - \bar{\Gamma}$ measured by 74 eV photons, overlaid with bulk density functional theory (DFT) calculations at the $k_z = 0$ and $\pi$ planes showing

the Kramers nodal line behavior. **d** Surface calculation for the $\bar{\Gamma} - \bar{M} - \bar{K} - \bar{\Gamma}$ projected onto the In termination is displayed for comparison, as the surface states from S termination have degraded for the angle-resolved photoemission spectroscopy (ARPES) data in (**c**). More details of the termination-dependent electronic band structure are elaborated in Supplementary Note 5. DFT calculations in comparison with ARPES data have no $E_F$ adjustment.

InTaS₂. The underlying quasiparticles could contribute to the properties of InTaS₂, for instance, the superconductivity.

## Discussion

The KNLs in noncentrosymmetric achiral little group have been clearly demonstrated in the exemplary case of the $P\bar{6}m2$ intercalated TMD superconductors consisting of In$_x$TaS₂ (*x* = 1/2, 1) and PbTaSe₂ (Supplementary Note 6), where In$_{1/2}$TaS₂ showcases the cleanest FS with a single KNL crossing the Fermi level, termed as the "ideal Kramers nodal line metal". The spin-orbital texture in In$_{1/2}$TaS₂ directly observed via spin-resolved ARPES offers a natural explanation for the spin-valley polarization engendered by the underlying broken inversion symmetry of the KNL little group, with a large SOC spin splitting up to around 250 meV. Furthermore, in InTaS₂, combined ARPES and angle-dependent QO studies point to the existence of the pinch points enforced by the KNLs, especially for the experimentally-probed closed $\epsilon$ pockets intersecting both the $\Gamma$–$M$ and $\Gamma$–$A$ KNLs. These pinch points are reminiscent of the 2D massless Dirac fermions on the surface of a 3D topological insulator, but reside on the intersection of the 3D FSs of an ideal KNL metal. Therefore, when the pinch points that harbor $n\pi$ ($n \in \mathbb{Z}$ and is an odd number) Berry phase are gapped by superconductivity, they produce nontrivial vortex spectra hosting chiral Majorana zero modes, which we elaborate in Supplementary Note 8 based on the model analysis of an ideal KNL. And in this work, both the KNLs with their associated pinch points and superconductivity are demonstrated in the In$_x$TaS₂ family.

More importantly, our work establishes an entire large family of quantum materials as a platform for realizing and tuning KNLs – intercalated TMD compounds. Specifically, the In site can be populated with different concentrations of Tl, Pb, Bi, and Sn, the Ta site with Nb, and the S site with Se[32]. The intercalation not only produces the broken inversion symmetry – a central requirement for realizing KNLs, but can also be utilized to introduce correlated physics and electronic orders in the presence of KNLs, such as magnetic orders, charge-density waves, or superconductivity, a regime that has not been previously explored. Therefore, this material family offers promising ingredients for spin and valley transport, axionic quasiparticles, and topological superconductivity.

Going beyond material selection, two other directions would deserve further exploration. The first one is the interplay between dimensionality and unconventional superconductivity. Recently, unconventional nodal superconductivity has been reported in monolayer 1*H*-TaS₂[51]. In the exfoliable intercalated $P\bar{6}m2$ TMD family represented by indium intercalated 1*H*-TaS₂ single crystals, the nature of the superconductivity deserves further investigation, especially upon approaching the few-layer limit. The second direction is strong electronic correlations. As analyzed in Supplementary Note 8, due to the Dirac physics of the KNL model, the AHE can be driven by Zeeman or exchange fields along the KNLs. However, the strength of the anomalous Hall conductivity is inversely proportional to the velocity of the KNL band. In the case of In$_x$TaS₂, the dispersive KNLs would give rise to an AHE that is overwhelmed by the ordinary Hall contributions.

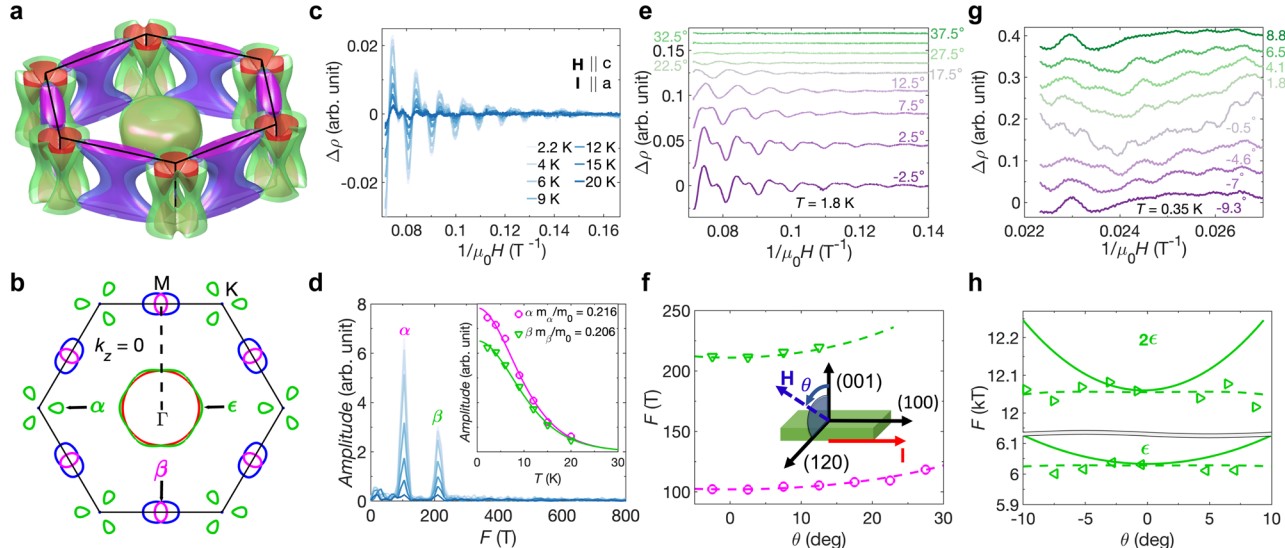

**Fig. 5 | Bulk Fermi surfaces and quantum oscillations of InTaS₂. a** Three-dimensional Fermi pockets of $InTaS_2$ calculated from first-principles. **b** The $k_z = 0$ slice of the Fermi pockets in panel (**a**), indicating the assignment of quantum oscillation frequencies $\alpha$, $\beta$, and $\epsilon$. **c** Temperature-dependent Shubnikov de-Haas (SdH) oscillations up to 14 T. **d** The fast Fourier transform of (**c**), illustrating the existence of two distinct frequencies. The inset is the fit to the Lifshitz–Kosevich thermal damping term to extract the effective masses fitting of the $\alpha$ and $\beta$ frequencies. **e** Angle-dependent SdH oscillations up to 14 T at 1.8 K. (**f**) $\alpha$ and $\beta$ frequencies as functions of the magnetic field orientations (symbols) along with

their values according to density functional theory (DFT) calculations (dashed lines). Inset: Sketch of the configuration used for the measurements. **g** Angle-dependent SdH oscillations from 40 T to 44.8 T at 0.35 K, illustrating the existence of higher frequencies. **h** Frequency of the $\epsilon$ cross-sectional area and its second harmonic as a function of the magnetic field orientations (symbols) including values from DFT calculations (dashed lines) and values from a cylinder (lines). The adjustment of the Fermi level of the DFT results to match the quantum oscillation data is elaborated in the Supplementary Note 7.

One way to increase the anomalous Hall signal is to induce flat KNLs at the Fermi level, which can be achieved in localized *f*-orbital electronic systems with one example in $P\bar{6}m2$ being UAuSi[56]. Such scenario may open an alternative avenue for topological phases and observables driven by strong correlations.

## Methods

### Crystal growth

Single crystals of $In_xTaS_2$ ($x$ = 1/2, 2/3, 1) were grown by the chemical vapor transport (CVT) method. The polycrystalline precursor of $InTaS_2$ was first prepared by a solid-state reaction. Indium powder, tantalum powder, and sulfur powder were mixed homogeneously with a mortar and pestle and sealed in quartz tubes under vacuum. The powder was heated up to 850℃ in 17 hours and kept at this temperature for 24 h before cooling down to room temperature. Approximately 3g of $InTaS_2$ and the transport agent $InCl_3$ (1mg/cm³ for $InTaS_2$, 4mg/cm³ for $In_{1/2}TaS_2$ and $In_{2/3}TaS_2$) were put together in a quartz tube (200 mm length, inner diameter 16 mm). All treatments were carried out in an argon box, with an oxygen and water content of less than 0.5 p.p.m. The quartz tubes were sealed and put into a two-zone furnace. The hot end with the starting materials was kept at 1050℃, and the cold end was kept at 1000℃. Single crystals of $In_xTaS_2$ ($x$ = 1/2, 2/3, 1) can be found in the middle of the quartz tube after a growth of 7 days. The single crystals of different compositions can be further distinguished by XRDs due to the differences in the lattice constant $c$.

### Transport and thermodynamic measurements

The magnetization measurements were performed in a Quantum Design Magnetic Property Measurement System-3 (MPMS-3) magnetometer with a He-3 option. The magnetotransport measurements up to 14 T were measured in a Quantum Design Physical Property Measurement System (PPMS) dynacool system equipped with a dilution refrigerator option. The resistance was measured through the ETO option. The magnetotransport measurements on the same sample were measured at the National High Magnetic Field Laboratory

(NHMFL) in Tallahassee in Cell-15, i.e, hybrid magnet, under magnetic field up to 44.8 T and temperature down to 0.35 K. The resistance was measured through a Lakeshore 370 AC resistance bridge.

### ARPES measurements

Angle-resolved photoemission spectroscopy (ARPES) and spin-ARPES measurements on $In_{1/2}TaS_2$ were collected at the Advanced Light Source, beamline 10.0.1.2 under *p* polarized photons, equipped with a Scienta Omicron DA30L spectrometer. The $In_{1/2}TaS_2$ samples were cleaved in situ with a base pressure better than $4 \times 10^{-11}$ Torr and at a maintained temperature of 15 K. During spin-ARPES measurements, VLEED (very low-energy electron diffraction) detectors were used with the spin quantization axis fixed along the out-of-plane ($s_z$) direction in Fig. 3 and the spin polarization is calculated from

$$P = \frac{1}{S}\frac{I_\uparrow - I_\downarrow}{I_\uparrow + I_\downarrow}, \quad (1)$$

where $S$ is the Sherman function. During the time of the experiment, the Sherman function took the value of 0.2. The corresponding spin-up and spin-down EDCs were measured up to the same acquisition time and normalized by the area using the background counts between 1.2 and 1.7 eV binding energies. The error bars of the spin polarization are calculated using the error propagation formula:

$$\delta P = P \cdot \sqrt{\frac{(\sqrt{I_\uparrow})^2 + (\sqrt{I_\downarrow})^2}{(I_\uparrow + I_\downarrow)^2} + \frac{(\sqrt{I_\uparrow})^2 + (\sqrt{I_\downarrow})^2}{(I_\uparrow - I_\downarrow)^2}}, \quad (2)$$

where the uncertainty of the spin-resolved photoelectron counts takes the form of $\sqrt{I_\uparrow}$ and $\sqrt{I_\downarrow}$ assuming the Poisson statistics of $I_\uparrow$ and $I_\downarrow$. In this calculation, the uncertainty from the Sherman function is not taken into consideration. The large error bar on one data point in Fig. 3e comes from the close values of spin up and spin down counts in the raw data. By definition, identical spin up and down counts give rise

to a divergent uncertainty of the spin polarization. Additionally, the positions of $k_3$ and $k_4$ in Fig. 3f were actually taken on the opposite sides of the $\bar{K} - \bar{M} - \bar{K}'$, closer to the $\bar{\Gamma}$ hexagonal pocket than the cut in Fig. 3a. We instead showed the cut to be outside of the BZ just for illustration purpose of a complete FS image. The curvature of the momentum cut on the FS in Fig. 3a, f is omitted to avoid curved and distorted band dispersions in the 3D plot.

ARPES data on $InTaS_2$ and $PbTaSe_2$ were taken at the Stanford Synchrotron Radiation Lightsource, Beamline 5-2 and the Brookhaven National Lab, National Synchrotron Light Source II, Beamline 21-ID, respectively. Both are equipped with a DA30 electron analyzer with vertical slit and have linear horizontal, linear vertical, and circularly polarized photons available. Only the data measured by linear horizontal light were included in the main text and the Supplementary Information. Samples of both kinds were cleaved in situ under a based pressure better than $3 \times 10^{-11}$ Torr and temperatures below 30 K. All ARPES measurements in the standard mode maintain an energy and angular resolution superior to 20 meV and 0.1°, while spin-ARPES on $In_{1/2}TaS_2$ has the energy and angular resolution better than 50 meV and 1°.

## Ab initio calculations

The Bloch spectral function and the one-step model ARPES calculations were carried out using the spin-polarized relativistic Korringa-Kohn-Rostoker (SPR-KKR) package[57], under the full-potential fully-relativistic four component Dirac formalism. Exchange-correlation potential within the local spin density approximation by Vosko, Wilk, and Nusair[58] was used. To model the random vacancy on the indium site in $In_xTaS_2$ ($x = 1/2, 2/3$), single-site coherent potential approximation was employed to obtain an auxiliary effective medium that reproduces the concentration-averaged scattering properties[57,59]. The KKR equations were solved with an angular momentum cutoff of $l_{max} = 4$ to account for the occupied 4f orbitals of Ta and the needs for spectroscopy calculations. The ARPES calculations considered a semi-infinite surface model terminated by the In atoms under the experimental geometry and used the layer-KKR multiple scattering theory, together with coherent potential approximation[60,61]. Therefore, such theory takes into account all factors such as light polarization, matrix element, final-state, surface, disorder, relativistic, and multiple scattering effects. Experimental crystal structures of $In_{1/2}TaS_2$ and $In_{2/3}TaS_2$ were utilized in the calculations[32,33].

For the calculations on $InTaS_2$, we used the Vienna Ab initio Simulation Package (VASP)[62] with the Perdew–Berke–Ernzerhof (PBE) exchange-correlation functional in the generalized-gradient approximation[63,64] to perform the density functional theory (DFT) calculations[65]. The crystallographic data were obtained from the topological material database[66–68]. SOC is included in the first-principles calculations. The Wannier tight binding model was further obtained through the Wannier90 package[69], which accurately fits the DFT bands with the inclusion of SOC. From this Wannier model, we studied in detail the topological crossings in $InTaS_2$. To better match with the ARPES data, we used a surface potential of −0.7 eV on the In $s$ orbitals for the In termination and +0.2 eV on the Ta $d$ orbitals for the S termination.

## Data availability

Data for this study are available in the main text and the Supplementary Information, or can be accessed on Zenodo[70]. Further data that support the findings of this study are available from the corresponding authors upon request.

## Code availability

Two ab-initio DFT-based packages were used in this study. The SPR-KKR package is freely available under the specific user license and can be downloaded following registration at https://www.ebert.cup.uni-muenchen.de/index.php/en/software-en. The VASP package can be purchased from https://www.vasp.at/.

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

## Acknowledgements

The authors thank Shiming Lei for the discussion at the initial stage of the project. This work was mainly supported by the Department of Defense, Air Force Office of Scientific Research under Grant No. FA9550-21-1-0343. This research used resources of the Advanced Light Source and the Stanford Synchrotron Radiation Lightsource, both U.S. Department of Energy, Office of Science User Facilities under Contracts No. DE-AC02-05CH11231 and No. DE-AC02-76SF00515, respectively. Part of the research also used resources of the National Synchrotron Light Source

II, a U.S. Department of Energy (DOE) Office of Science User Facility operated for the DOE Office of Science by Brookhaven National Laboratory under Contract No. DE-SC0012704. The ARPES work at Rice University was also supported by the Gordon and Betty Moore Foundation's EPiQS Initiative through grant No. GBMF9470 and the Robert A. Welch Foundation Grant No. C-2175 (M.Y.). Y. Gao's work at the National High Magnetic Field Laboratory was funded by the Gordon and Betty Moore Foundation's EPIQS Initiative through ICAM-I2CAM, Grant GBMF5305. L.B. is supported by the US-DoE, BES program, through award DE-SC0002613. A portion of this work was performed at the National High Magnetic Field Laboratory, which is supported by National Science Foundation Cooperative Agreement No. DMR-2128556 and the State of Florida. Work at the University of California, Berkeley and Lawrence Berkeley National Laboratory was funded by the U.S. DOE, Office of Science, Office of Basic Science, Materials Sciences and Engineering Division under Contract No. DE-AC02-05CH11231 (Quantum Materials Program KC2202). Y.Z. acknowledges support from the US National Science Foundation (NSF) Grant Number 2201516 under the Accelnet program of Office of International Science and Engineering (OISE). The KKR calculation work was supported by the project Quantum materials for applications in sustainable technologies (QM4ST), funded as project No. CZ.02.01.01/00/22_008/0004572 by P JAK, call Excellent Research. K.T.L. acknowledges the support of the Hong Kong Research Grant Council through Grants RFS2021-6S03, C6025-19G, C6053-23G, AoE/P- 701/20, 16307622, 16309223 and 16311424. Research at UBC was undertaken thanks in part to funding from the Canada First Research Excellence Fund, Quantum Materials and Future Technologies Program. P.H. acknowledges funding from NSF grant No. DMR 2047193. M.H. and D.L. acknowledge the support of the U.S. Department of Energy, Office of Science, Office of Basic Energy Sciences, Division of Material Sciences and Engineering, under Contract No. DE-AC02-76SF00515.

## Author contributions

M.Y. and E.M. conceived the project. J.H. suggested the material family. Y.Gao synthesized the crystals and performed magnetotransport and magnetization measurements with the help from M.O., A.M.H., A.Moon, and L.B. under the supervision of E.M.. Y.Z., Y.Guo, Z.Y., and J.S.O. performed the ARPES experiments under the supervision of M.Y., R.J.B. and J.K., with the help from A.F., S.K.M., M.H., D.L., A.R., and E.V.. A.P. and Y.Z. performed the KKR calculations, with support from A.Marmodoro, under the supervision of J.M.. X.G. carried out the wave-function based first-principles calculations, with the help from X.J.G., under the supervision of K.T.L.. P.H. theoretically studied topological consequences of general KNLs. Y.Z., Y.Gao, and A.P. contributed equally to this work. All authors contributed to the manuscript preparation.

## Competing interests

The authors declare no competing interests.

## Additional information

[1]Department of Physics and Astronomy, Rice University, Houston, TX 77005, USA. [2]New Technologies Research Center, University of West Bohemia, Plzen 301 00, Czech Republic. [3]Department of Physics, Hong Kong University of Science and Technology, Clear Water Bay, Hong Kong, China. [4]Applied Physics Graduate Program, Smalley-Curl Institute, Rice University, Houston, TX 77005, USA. [5]Department of Physics, University of California, Berkeley, Berkeley, CA 94720, USA. [6]National High Magnetic Field Laboratory, Tallahassee, Tallahassee, FL 32310, USA. [7]Physics Department, Florida State University, Tallahassee, FL 32306, USA. [8]Stewart Blusson Quantum Matter Institute, University of British Columbia, Vancouver, Vancouver, BC V6T 1Z4, Canada. [9]Advanced Light Source, Lawrence Berkeley National Laboratory, Berkeley, Berkeley, CA 94720, USA. [10]Stanford Synchrotron Radiation Lightsource, SLAC National Accelerator Laboratory, 2575 Sand Hill Road, Menlo Park, CA 94025, USA. [11]National Synchrotron Light Source II, Brookhaven National Lab, Upton, NY 11973, USA. [12]Rice Center for Quantum Materials, Rice University, Houston, TX 77005, USA. [13]Department of Electrical and Computer Engineering, Rice University, Houston, TX 77005, USA. [14]Department of Materials Science and NanoEngineering, Rice University, Houston, TX 77005, USA. [15]Smalley-Curl Institute, Rice University, Houston, TX 77005, USA. [16]Department of Physics & Astronomy, University of British Columbia, Vancouver, Vancouver, BC V6T 1Z1, Canada. [17]Canadian Institute for Advanced Research, Toronto, Toronto, ON M5G 1M1, Canada. [18]Materials Science Division, Lawrence Berkeley National Laboratory, Berkeley, Berkeley, CA 94720, USA. [19]Department of Physics and Texas Center for Superconductivity, University of Houston, Houston, TX 77204, USA. [20]These authors contributed equally: Yichen Zhang, Yuxiang Gao, Aki Pulkkinen. ✉e-mail: emorosan@rice.edu; mingyi@rice.edu

