## [Transparent Peer Review file · Nature Communications]

Kramers nodal lines in intercalated TaS₂ superconductors

Corresponding Author: Dr Ming Yi

Version 0:

Reviewer comments:

Reviewer #1

(Remarks to the Author)

Zhang et al. report the experimental observation of an ideal Kramer nodal lines (KNL) within the In_xTaS₂ material family. This system features the isolated KNL on the Fermi surface due to certain crystal symmetries and strong spin-orbital coupling effect. Using high-resolution and spin-resolved Angle-Resolved Photoemission Spectroscopy (ARPES), they were able to directly visualize these bands, including their spin characteristics. These results align well with theoretical predictions. In particular, for InTaS₂ (where x=1), which exhibits relatively complex 3D Fermi surfaces, quantum oscillation (QO) study was further employed to identify the specific band components from the KNL.

Identifying ideal and clear topological quantum states, such as the KNL observed here, is important in the field, as it elucidates the topological-related behaviors of the system. Furthermore, the persistent superconductivity observed within this material family, offers additional opportunity to study the interplay between topological states and strong correlations. The data and theoretical analysis are of high quality and the conclusions are well supported by the results. I would recommend publication of the manuscript after the following technical/presentation issues can be addressed to enhance the overall readability.

1. In the case of In_{1/2}TaS₂, the authors note that “k_z = π dispersions show the most prominent splitting.” Similarly, for InTaS₂, the authors state “InTaS₂ is more three-dimensional and exhibits stronger k_z band dispersions”. Including photon-energy-dependent ARPES data would enhance the understanding of the k_z contributions in these cases.
2. When addressing ARPES results for InTaS₂, calculations for both In and S terminations are summed together. It would be better to present these two sets of calculations separately in the SI for clarity. Additionally, I am expecting this termination dependence to also apply to other compounds in the In_xTaS₂ family, for instance the In_{1/2}TaS₂, can the authors further elaborate?
3. Theoretically the “random vacancy modelling on the In sites” was used, I am curious about the overall spatial homogeneity of the In_xTaS₂ crystals - the In intercalation. Some real-space scanning, for example, the In concentration may help with the understanding.
4. The introduction part of the manuscript, also Fig. 1, emphasizes the In_xTaS₂ (x=1/2, 2/3, 1) family compounds. However, the electronic structure of x=2/3 is not discussed at all in the main text. PtTaSe₂ is mentioned instead but seemingly irrelevant to the discussion.
5. I find the current Figure 1 somewhat distracting. Some suggestions include: (1) Highlight the specific symmetries crucial to form the KNL in the crystal structure illustration. (2) emphasis the KNL in Fig. 1d with clearer notations. The current black lines are nearly invisible. (3) Using the relatively simpler Fermi surfaces of In_{1/2}TaS₂ to showcase the key point of the study- the KNL-while moving the current Fig 1e of InTaS₂ to Fig. 4 (or alongside QO results to Fig. 5), where it is mostly discussed.

Reviewer #2

(Remarks to the Author)

The authors present a paper focused on Kramers nodal lines (KNLs) found in intercalated In-TaS₂. This work explores the presence of KNLs in a class of noncentrosymmetric achiral superconductors. The authors establish that these materials host ideal KNL phases, which are unique topological electronic structures that emerge due to broken inversion symmetry. The evidence is provided through angle-resolved photoemission spectroscopy (ARPES), spin-resolved ARPES, quantum oscillations, and first-principles calculations, which together offer direct confirmation of KNLs well-isolated at the Fermi level. The authors claim that these findings reveal strong Ising-type spin-orbit coupling (SOC), spin-valley polarization, and

potential applications in spintronics and valleytronics. Additionally, they discuss the interplay between KNLs and superconductivity, proposing implications for topological superconductivity and quantum transport phenomena. This paper represents an advancement in the field, with claims that are well-supported by the data. I believe it aligns well with the scope of Nature Communications, and based on my expertise, I recommend that the work be accepted with minimal changes.

Reviewer #3

(Remarks to the Author)

The manuscript by Yichen Zhang et al. identifies the InTaS materials featuring Kramers nodal lines (KNLs) located near/on the Fermi level, as confirmed by both ARPES experiments and DFT calculations. The authors define these as "ideal KNLs". For the specific composition In_xTaS with $x=1/2$, the authors primarily examine the spin degeneracy of the KNL and its location in momentum space using spin-integrated ARPES. Additionally, the authors employ spin-resolved ARPES to reveal Ising-type spin-valley locking near the H point. These experimental findings are consistently supported by DFT calculations. The authors further investigate the Fermi surfaces of In_xTaS with $x=1$ through quantum oscillation measurements.

The manuscript is written well and provides a thorough exploration of intercalated TaS₂, presenting a wealth of data, including resistivity measurements to study superconductivity, spin-integrated and spin-resolved ARPES to characterize KNLs and spin-valley polarization, as well as theoretical calculations of band structures, spin polarization, and analysis of vortex topology induced by KNLs.

As the first study to identify a material exhibiting ideal KNLs and to provide such a comprehensive investigation, the manuscript is suitable for publication in Nature Communications. However, I have a few comments and suggestions for improvement:

(1)The authors state, "the momentum dependent relativistic pseudospin splitting pattern is reminiscent of the recently generalized concept of nonrelativistic altermagnetism". It should be careful about this statement since the KNL does not break time-reversal symmetry, but altermagnet does.

(2)The connection between spin-valley polarization and KNLs is not sufficiently clear in the manuscript. While KNLs arise from non-symmorphic symmetries, spin-valley polarization typically results from spin-orbit coupling. The authors should provide a more detailed explanation of how KNLs contribute to or interact with spin-valley polarization. Specifically, does the existence of KNLs play a direct role in generating spin-valley polarization, or are these phenomena independently driven by different mechanisms? Further elaboration on this relationship would strengthen the manuscript.

(3)The experiment investigation of KNL and spin-valley is mainly focused on In_xTaS₂ with $x=1/2$, while the quantum oscillation measurement is conducted on In_xTaS₂ with $x=1$. There seems exist a discrepancy. To me, the role of KNLs in quantum oscillations, and potentially in magnetic breakdown, is an intriguing aspect that warrants further exploration. Given that In_xTaS₂ with $x=1/2$ exhibits the best (quasi-)two-dimensional band structure among the studied samples, it is surprising that the authors did not investigate quantum oscillations in this material. Such measurements could provide valuable insights into the Fermi surface topology and the influence of KNLs on electronic properties.

(4)Given that the authors successfully fit the ARPES data using DFT calculations, it would be highly beneficial to construct a tight-binding model that captures the essential features of the bands near the Fermi level, including the Kramers nodal lines (KNLs) and spin-valley polarization. For example, the Figure for KNL is done by toy model in Fig. 1d. Such a model would serve as a valuable starting point for future studies, particularly in exploring the interplay between superconductivity and the topological features of the material.

Version 1:

Reviewer comments:

Reviewer #1

(Remarks to the Author)

The authors have addressed all the comments well and made appropriate revisions accordingly. I recommend this manuscript for publication in Nature Communications in its current form.

Reviewer #2

(Remarks to the Author)

N/A

Reviewer #3

(Remarks to the Author)

All the comments are addressed, so I suggest the current form published in Nat Comm.

Reviewer 1

Zhang et al. report the experimental observation of an ideal Kramer nodal lines (KNL) within the In_xTaS_2 material family. This system features the isolated KNL on the Fermi surface due to certain crystal symmetries and strong spin-orbital coupling effect. Using high-resolution and spin-resolved Angle-Resolved Photoemission Spectroscopy (ARPES), they were able to directly visualize these bands, including their spin characteristics. These results align well with theoretical predictions. In particular, for InTaS_2 (where $x=1$), which exhibits relatively complex 3D Fermi surfaces, quantum oscillation (QO) study was further employed to identify the specific band components from the KNL.

Identifying ideal and clear topological quantum states, such as the KNL observed here, is important in the field, as it elucidates the topological-related behaviors of the system. Furthermore, the persistent superconductivity observed within this material family, offers additional opportunity to study the interplay between topological states and strong correlations. The data and theoretical analysis are of high quality and the conclusions are well supported by the results. I would recommend publication of the manuscript after the following technical/presentation issues can be addressed to enhance the overall readability.

Reply: We are thankful to the reviewer for the nice summary of our manuscript and the technical/presentation suggestions to improve our manuscript, as well as the recommendation for publication.

1. In the case of $\text{In}_{1/2}\text{TaS}_2$, the authors note that “ $k_z = \pi$ dispersions show the most prominent splitting.” Similarly, for InTaS_2 , the authors state “ InTaS_2 is more three-dimensional and exhibits stronger k_z band dispersions”. Including photon-energy-dependent ARPES data would enhance the understanding of the k_z contributions in these cases.

Reply: We thank the reviewer for this suggestion of including photon-energy-dependent ARPES data that would enhance the understanding of the k_z contributions in the ARPES spectra. We completely agree and have inserted a new Supplementary Note 4 with Supplementary Fig. 4 and Supplementary Fig. 5 to discuss in detail the ARPES photon-energy-dependence of both compounds in the vacuum ultraviolet regime. The summary of the added contents in a nutshell is that for $\text{In}_{1/2}\text{TaS}_2$, we identify a k_z -dispersive band feature near 1 eV below the Fermi level, consistent with the density functional theory calculations, hence revealing an inner potential of

around 13 eV. Based on such analysis, the 75 eV photon energy used to probe ARPES and spin-ARPES spectra of $\text{In}_{1/2}\text{TaS}_2$ in the main text Fig. 2 and Fig. 3 is located close to the $k_z = 0$ plane. However, for InTaS_2 due to the strong k_z broadening effect mentioned in the main text, the ARPES spectra are highly k_z -integrated within the measured photon energy range, hindering a clear determination of the k_z positions. This was also one of the motivations of resorting to the bulk-sensitive quantum oscillation probe.

Below, we reproduce the new Supplementary Note 4 to answer the question of photon-energy-dependence of ARPES in $\text{In}_{1/2}\text{TaS}_2$ and InTaS_2 .

***** Reproduced contents start here *****

Supplementary Figure 4: Photon-energy-dependent electronic band dispersions of In_xTaS_2 measured by ARPES. (a-d) Photon-energy-dependent k_x - k_z mappings of In_xTaS_2 at binding energies of 0.02, 0.20, 1.00, and 1.20 eV, respectively. k_z corresponds to the varying photon energy and k_x is along the $\bar{\Gamma} - \bar{M}$ high symmetry direction. The two red arcs in (c) correspond to the two photon energies utilized in (f) and (g) for the band dispersions, with the solid half being ARPES data and the dashed half replaced by DFT results at the designated constant k_z . The black dashed curve of 75 eV corresponds to the photon energy used for the data in main text Figs. 2 and 3. (e) $E - k_z$ band dispersions extracted along the “V cut” direction indicated by the vertical red arrow in panel (c). The direction contains $\bar{\Gamma} - A$, of which the DFT result is provided with on the right. (f) and (g) ARPES band structure along $\bar{\Gamma} - \bar{M}$ ($k_z \approx 0$) and $A - L$ ($k_z \approx \pi$) at 78 and 96 eV, respectively, in comparison with the DFT results. The red dashed boxes denote the distinct features at around 1 eV binding energy for the two different photon energies. All data in this figure were acquired around 20 K. An inner potential of 13 eV was experimentally adopted for the $h\nu$ to k_z conversion.

Both $\text{In}_{1/2}\text{TaS}_2$ and InTaS_2 are quasi-two-dimensional (2D) materials with the building blocks of In-layer and TaS_2 -layer. One would expect that the two materials exhibit weak photon-energy-dependence of the band dispersions in the vacuum ultraviolet (VUV) regime of incident photons, complicated by kinetic-energy-dependent matrix element effects that vary with the photon energy. To clarify their photon-energy-dependence and have a rough estimate of the k_z positions of the acquired data, we present in Supplementary Fig. 4 and Supplementary Fig. 5 the ARPES $h\nu$ -dependence of $\text{In}_{1/2}\text{TaS}_2$ and InTaS_2 , respectively. We note that one can estimate the k_z resolution of VUV-ARPES using the inelastic mean free path of photoelectrons λ_{IMFP} : $\Delta k_z \approx \hbar/\lambda_{IMFP}$, where \hbar is the reduced Planck constant. Typically for VUV photons, Δk_z is around the order of 0.1 \AA^{-1} . Therefore, we emphasize that our results only provide a rough estimate of the k_z dispersions in these layered materials.

As shown in Supplementary Figs. 4(a) and (b), the isolated Kramers nodal line (KNL) band along $\bar{\Gamma} - \bar{M}$ in In_xTaS_2 shows two straight line-like features along k_z near the Fermi level, indicating that the Fermi surface is highly 2D, consistent with the 3D voxel-style Fermi surface topology of $\text{In}_{1/2}\text{TaS}_2$ calculated in the main text Fig. 1(e). The k_z dispersionless feature is also facilitated by the surface-sensitivity of VUV APRES. We note that the $\text{In}_{1/2}\text{TaS}_2$ samples have doping variations even within the same batch of synthesis. Therefore, the x of In concentration could vary near 0.5. From the hole-doping trend of the band positions in Supplementary Fig. 4, in comparison with the main text Figs. 2 and 3, we speculate that this sample has a slightly lower concentration of In, but also close to $x = 0.5$. Furthermore, upon examining the E - k_x - k_z ARPES data at deeper binding energies of 1 and 1.2 eV, as shown in Supplementary Figs. 4(c) and (d), more k_z dispersive features show up. To determine the inner potential for the photon energy $h\nu$ to k_z conversion, we examine the E - k_z band dispersions in Supplementary Fig. 4(e), taken from the vertical (V) cut in Supplementary Fig. 4(c), in comparison with the Korringa-Kohn-Rostoker density functional theory (KKR-DFT) results along $\Gamma - A$. It is clear in the DFT prediction that a k_z -dispersive band reaches its band-top around 1 eV binding energy at A ($k_z = \pi$), accompanied by the non- k_z -dispersive bands at around $E - E_F = -2$ eV. Therefore, we interpreted our photon-energy-dependent ARPES data with an inner potential choice of 13 eV, so that the k_z dispersive intensities at high photon energies (higher k_z) in Supplementary Fig. 4(e) are centered around the Γ near $k_z = 5.5 \text{ \AA}^{-1}$ and match with the characteristic of band-top appearing at A . Such Γ -symmetric feature is also denoted by the two red horizontal arrows in Supplementary Fig. 4(d) for the k_x - k_z constant energy contour at $E - E_F = -1.2$ eV. Notice that the higher the photon energy is, the better the k_z

resolution is. Therefore, data at lower photon energy (lower k_z) can be poorly resolved, while modulated by photoemission matrix element effects. Despite such complications, we note that the k_z -dispersive band still reaches its top for another A near $k_z = 4.3 \text{ \AA}^{-1}$ (see Supplementary Fig. 4(e)), under the 13 eV inner potential. Further, we point out that the non- k_z -dispersive bands predicted by DFT at around 2 eV binding energy along $\Gamma - A$ are enhanced for the ARPES data between $k_z = 4.4$ and 5.1 \AA^{-1} in Supplementary Fig. 4(e), likely due to the kinetic-energy-dependent photoemission matrix element affected by the final-state effects. In Supplementary Figs. 4(f) and (g), we subsequently showcase the E - k_x band dispersions corresponding to the 78 and 96 eV indicated in Supplementary Fig. 4(c), respectively, while half of the ARPES data are replaced by the DFT calculations at the correspondingly constant k_z for an in-depth comparison. Importantly, the red dashed boxes in Supplementary Figs. 4(f) and (g) emphasize the appearance of the A band-top near 1 eV binding energy at 96 eV (near $k_z = \pi$) but its disappearance at 78 eV (near $k_z = 0$). Therefore, the 75 eV photon energy employed in the main text Figs. 2 and 3 correspond to a k_z position near $k_z = 0$, close to the k_z plane of the bulk KNLs.

Supplementary Figure 5: Photon-energy-dependent ARPES band structure of InTaS_2 . (a) and (b) In-plane Fermi surface contours of InTaS_2 measured by 74 and 62 eV photons. (c1)-(c7) $\Gamma - M$ band dispersions obtained with a sequence of increasing photon energies: 45, 55, 65, 74, 85, 95, and 105 eV. The momentum

range of (c1)-(c7) is also indicated by the vertical red dashed arrow in panel (a). Light polarization is the same with that of the main text Fig. 4. The measurement temperature is around 23.5 K.

Further, we examine the ARPES photon-energy-dependence of the stoichiometric InTaS₂. Two k_x - k_y in-plane Fermi surfaces are shown in Supplementary Fig. 5(a) and Supplementary Fig. 5(b). Despite the different photon energies of 74 eV and 62 eV, they show similar band features as calculated in main text Fig. 4(b), but with certain intensity modulation due to the kinetic-energy-dependent matrix element effects. Subsequently, we focus on the $\bar{\Gamma} - \bar{M}$ cut spanned in Supplementary Fig. 5(a) and vary the photon energy in an approximated step of 10 eV, as shown in Supplementary Figs. 5(c1)-(c7). Based on the assumption of a 15 eV inner potential and the c lattice constant of InTaS₂, a 10 eV step in our scanned photon energy range would roughly correspond to a change from $k_z = 0$ to $k_z = \pi$. However, as shown in Supplementary Fig. 5(c), we do not observe clear k_z dispersions within the measured binding energy range. We attribute this to the strong k_z broadening effect in InTaS₂, which is also one of the motivations for the bulk-sensitive quantum oscillation measurements on InTaS₂ presented in main text Fig. 5.

***** **Reproduced contents end here** *****

2. When addressing ARPES results for InTaS₂, calculations for both In and S terminations are summed together. It would be better to present these two sets of calculations separately in the SI for clarity. Additionally, I am expecting this termination dependence to also apply to other compounds in the In_xTaS₂ family, for instance the In_{1/2}TaS₂, can the authors further elaborate?

Reply: We thank the reviewer very much for the comments about the termination dependence of InTaS₂ and In_{1/2}TaS₂. It would indeed be better to present the In and S terminations separately in the Supplementary Information (SI) for clarity. Therefore, we have included a new Supplementary Note 5 in SI for discussing the termination dependence of both InTaS₂ and In_{1/2}TaS₂. The quick summary is that for InTaS₂, we clearly identify both experimentally and theoretically the separate contributions from In and S terminations, while for In_{1/2}TaS₂, both terminations are in principle possible, but we have only unambiguously observed the In/Vc termination in ARPES experiments. Despite that, we provide predictions on the S-terminated electronic band structure based on the state-of-the-art all-electron full-potential fully-relativistic one-step model ARPES calculations. Such predictions of the S-terminated band structure potentially explain the difficulty of identifying S termination in In_{1/2}TaS₂ ARPES experiments, due to the effort required to

distinguish from extrinsic cleaving domain effect which typically arises from a finite experimental beam spot. To provide a more concrete answer to the termination-dependent ARPES spectra of In_xTaS_2 , we reproduce the newly inserted Supplementary Note 5, and Supplementary Figs. 6 and 7 below.

***** Reproduced contents start here *****

Supplementary Figure 6: Termination-dependent electronic band structure of InTaS_2 (a) Fermi surface (FS) measured by 74 eV photons, reproduced from the main text Fig. 4(a). (b1), (b2), (b3) DFT-based (after Wannierization) surface calculations of the FS with summed In and S terminations, In termination, and S termination, respectively. (c) Band dispersions along $\bar{\Gamma} - \bar{M}$ within the same measurement shown in panel (a), with co-existing In and S terminations under the beam spot. (d) DFT-based surface calculations for the $\bar{\Gamma} - \bar{M}$ band structure projected to the In termination. (e) Same as (d) but projected to the S termination. ISS: indium surface states. SSS: sulfur surface states. Calculation results in (d) and (e) are multiplied by a Fermi-Dirac distribution function at 23.5 K convolved with a hypothetical experimental energy resolution of 20 meV at the Fermi level. All calculations presented here have no E_F adjustment.

Since the crystal structure of InTaS_2 allows two equally possible cleaving terminations during ARPES measurements, in the main text, we compare the ARPES data with a summation of In and S terminations on the Fermi surface, as reproduced by Supplementary Fig. 6(a) and Supplementary Fig. 6(b1). Furthermore, we decompose the Fermi surface (FS) contributions from the two terminations in Supplementary Fig. 6(b2) and Supplementary Fig. 6(b3). As can be seen from Supplementary Fig. 6(b2), the In termination contributes to the arc-like surface states

connecting the \bar{K} and \bar{K}' points across the $\bar{\Gamma} - \bar{M}$ direction, denoted by the red arrow and the “ISS” label. For the S termination, the surface states manifest as a pair of spin-orbital-split hexagonal pockets enclosing the broadened intensities centered around $\bar{\Gamma}$ and the trefoil-like pockets centered around \bar{K} (\bar{K}') points. They are denoted by the two blue arrows and the label “SSS”. The termination-dependent band structure along $\bar{\Gamma} - \bar{M}$ is then elaborated in Supplementary Figs. 6(c)-(e), devoted to ARPES experimental data with co-existing In + S terminations, theoretical pure In termination, and theoretical pure S termination, respectively. It is evident again the results in Supplementary Fig. 6(c) can be best described by a summation of the results in Supplementary Figs. 6(d) and (e). The “ISS” in Supplementary Fig. 6(d) gives out the arc-like feature in Supplementary Fig. 6(b2) at the FS, while the “SSS” band in Supplementary Fig. 6(e) manifests as the pinch point along $\bar{\Gamma} - \bar{M}$ on the hexagonal Fermi pockets enclosing $\bar{\Gamma}$ in Supplementary Fig. 6(b3). Notice that the “SSS”-band presented in Supplementary Fig. 6(e) is split from the bulk continuum, echoing the two-peak feature along $\bar{\Gamma} - \bar{M}$ predicted by the one-step model for the S-terminated $\text{In}_{1/2}\text{TaS}_2$ in Supplementary Fig. 7(c).

Supplementary Figure 7: Termination-dependent electronic band structure of $\text{In}_{1/2}\text{TaS}_2$ (a) Band dispersions measured by 75 eV p-polarized photons along $\bar{\Gamma} - \bar{M} - \bar{K} - \bar{\Gamma}$, reproduced from the main text Fig. 2(f). All red curves on top are the momentum distributions curves (MDCs) integrated from E_F to $E_F - 60$ meV along the corresponding high symmetry momentum directions indicated at the bottom. (b) One-step ARPES calculations using the In/Vc termination, reproduced from main text. 2(g), but with the integrated MDCs attached on top. Vc: vacancy. (c) One-step ARPES calculations using the same experimental conditions but with the S termination instead. The vertical red arrows between the MDCs and the band dispersion images are indicators for the one-peak or two-peak structure. All presented one-step model calculated data have a Fermi level adjustment of +75 meV, consistent with the operation in main text Fig. 2, and are applied with a Fermi-Dirac distribution function at 15 K, which is additionally convolved with a hypothetical experimental energy resolution of 20 meV at the Fermi level.

In terms of $\text{In}_{1/2}\text{TaS}_2$, it is in principle possible to have both In/Vc (Vc: vacancy) and S terminations, similar to the case of InTaS_2 . However, we do not have unambiguous experimental observation of the S termination and have only obtained high-quality

ARPES data on the In/Vc termination, of which the band dispersions along $\bar{\Gamma} - \bar{M} - \bar{K} - \bar{\Gamma}$, are reproduced in Supplementary Fig. 7(a). To quantitatively compare the fine structure near the Fermi level between experiment and theory, we present the momentum distribution curves (MDCs) integrated between E_F and $E_F - 60$ meV on top in arbitrary unit. The experimental data along $\bar{\Gamma} - \bar{M}$ clearly shows a one-peak structure in Supplementary Fig. 7(a). This is in excellent agreement with the results predicted by the one-step model of photoemission, shown in Supplementary Fig. 7(b), adopting a In/Vc termination. In addition to the consistent one-peak structure along $\bar{\Gamma} - \bar{M}$, the two-peak structure along $\bar{M} - \bar{K}$ in Supplementary Figs. 7(a) and (b) exhibits the same line profile where the right peak is higher than the left peak. These results are in sharp contrast with the one-step model prediction in Supplementary Fig. 7(c) adopting a S termination, where a double peak splitting is observed along $\bar{\Gamma} - \bar{M}$ and along $\bar{M} - \bar{K}$ the right peak is lower than the left peak under the same input of experimental measurement geometry. Therefore, we conclude that our experimentally probed termination is the In/Vc termination. We acknowledge that the matching between Supplementary Fig. 7(a) Supplementary Fig. 7(b) is not perfect, as the $\bar{K} - \bar{\Gamma}$ MDC line profile cannot be exactly reproduced by either Supplementary Fig. 7(b) or Supplementary Fig. 7(c). Experimentally, it is not straightforward to directly identify the S termination through the double-peak feature along $\bar{\Gamma} - \bar{M}$, since one must carefully distinguish this from band splitting caused by potential cleaving domains, because the material does not usually give a large area of single cleaving domain. Future experiments with smaller beam spot, combined with sulfur core-level spectroscopy study can further clarify and confirm the S termination scenario of $\text{In}_{1/2}\text{TaS}_2$.

***** **Reproduced contents end here** *****

3. Theoretically the “random vacancy modelling on the In sites” was used, I am curious about the overall spatial homogeneity of the In_xTaS_2 crystals - the In intercalation. Some real-space scanning, for example, the In concentration may help with the understanding.

Reply: We thank the reviewer for suggesting real-space scanning on the In_xTaS_2 samples with random site disorder. To ensure the consistency among different experimental techniques, we exfoliated from the same $\text{In}_{1/2}\text{TaS}_2$ sample measured by ARPES (main text Fig. 2) and spin-ARPES (main text Fig. 3) and performed scanning electron microscopy with energy dispersive x-ray spectroscopy (SEM-EDX) experiments on the exfoliated piece. The scan result shows that In presence is

homogenous across the entire scanned region, as well as for Ta and S. Additionally, the one-dimensional EDX scan at different regions of the sample showed a consistent atomic composition of In:Ta:S \approx 0.53:0.99:2. For this purpose, we inserted a new Supplementary Note 3 in the supplemental information. Below we reproduce the Supplementary Fig. 3 presented in Supplementary Note 3.

***** Reproduced contents start here *****

Supplementary Figure 3: Real-space scanning electron microscopy with energy dispersive x-ray spectroscopy (SEM-EDX) results of $In_{1/2}TaS_2$ at room temperature. (a-c) Spatial scan for the x-ray emitted by electrons returned to the L shell of In, the M shell of Ta, and the K shell of S, respectively. (d) Area-averaged EDX spectrum to determine the elemental composition of the nominal $In_{1/2}TaS_2$, yielding $In_{0.53}Ta_{0.99}S_2$.

***** Reproduced contents end here *****

4. The introduction part of the manuscript, also Fig. 1, emphasizes the In_xTaS_2 ($x=1/2, 2/3, 1$) family compounds. However, the electronic structure of $x=2/3$ is not

discussed at all in the main text. PtTaSe₂ is mentioned instead but seemingly irrelevant to the discussion.

Reply: We thank the reviewer for the pointing out this incoherence. In the revised version of the main text, we first only emphasize In_xTaS₂ (x=1/2 and 1), since this is the experimental focus of the main figures. Therefore, we took off the x=2/3 text label in main text Fig. 1(b), leaving only the x=1/2 label which corresponds exactly to the illustrated crystal structure. Then in the “Introduction” part, we modified the main text to be “These are exemplified by In_xTaS₂ (x=1/2 and 1) (Fig. 1), both of which also feature superconducting ground states³⁴⁻³⁶.” Then to demonstrate the generality of the KNLs physics in this superconducting material family, we followed the modified statement with “In addition, similar physics of the KNLs in the isostructural intercalated TMD superconductors, In_{2/3}TaS₂ and PbTaSe₂, is discussed in the supplemental information (notes 2 and 6)”, pointing interested readers towards further information.

5. I find the current Figure 1 somewhat distracting. Some suggestions include: (1) Highlight the specific symmetries crucial to form the KNL in the crystal structure illustration. (2) emphasis the KNL in Fig. 1d with clearer notations. The current black lines are nearly invisible. (3) Using the relatively simpler Fermi surfaces of In_{1/2}TaS₂ to showcase the key point of the study-the KNL-while moving the current Fig 1e of InTaS₂ to Fig. 4 (or alongside QO results to Fig. 5), where it is mostly discussed.

Reply: We appreciate the constructive suggestions from the reviewer on improving the structure of the manuscript. (1) We highlight the most important symmetry operation for the formation of the KNLs in In_xTaS₂, the horizontal mirror M_z within the Ta atomic layer. Now this feature is the half-transparent magenta planes in the updated main text Fig. 1(b). Further we emphasized the inversion symmetry breaking by adding texts in Fig. 1(b) and in contrast outlined the preserved inversion center in 2H-TaS₂ in Fig. 1(a). (2) To emphasize the KNLs in Fig. 1(d), we slightly rotated the orientation of the 3D illustration and mark the KNLs with the more contrasted green tubes. An additional green arrow was pointed to the KNL tubes to denote “KNLs”. (3) We have replaced the Fermi surfaces of InTaS₂ with the relatively simpler Fermi surfaces of In_{1/2}TaS₂ in the updated Fig. 1(e), where both a voxel-style 3D and 2D Fermi surfaces based on first-principles calculated Bloch spectral function intensities were presented. The 2D Fermi surface slices at k_z = 0 and k_z = π further showcase the Fermi surface pinch points along Γ – M and A – L for the pair of pockets enclosing Γ – A. In accordance with such changes, we split

the original Fig. 4 into the new Fig. 4 and Fig. 5, where the original Fig. 1(e) showing the 3D Fermi surface and 2D $k_z = 0$ sliced Fermi surface of InTaS₂ was moved to Fig. 5(a) and (b) alongside the quantum oscillation results, as suggested. The descriptions in the main text are also modified correspondingly.

Reviewer 2

The authors present a paper focused on Kramers nodal lines (KNLs) found in intercalated In-TaS₂. This work explores the presence of KNLs in a class of noncentrosymmetric achiral superconductors. The authors establish that these materials host ideal KNL phases, which are unique topological electronic structures that emerge due to broken inversion symmetry.

The evidence is provided through angle-resolved photoemission spectroscopy (ARPES), spin-resolved ARPES, quantum oscillations, and first-principles calculations, which together offer direct confirmation of KNLs well-isolated at the Fermi level. The authors claim that these findings reveal strong Ising-type spin-orbit coupling (SOC), spin-valley polarization, and potential applications in spintronics and valleytronics. Additionally, they discuss the interplay between KNLs and superconductivity, proposing implications for topological superconductivity and quantum transport phenomena.

This paper represents an advancement in the field, with claims that are well-supported by the data. I believe it aligns well with the scope of Nature Communications, and based on my expertise, I recommend that the work be accepted with minimal changes.

Reply: We thank the reviewer very much for the nice summary of our work and the recommendation of the acceptance at Nature Communications with minimal changes.

Reviewer 3

The manuscript by Yichen Zhang et al. identifies the InTaS materials featuring Kramers nodal lines (KNLs) located near/on the Fermi level, as confirmed by both ARPES experiments and DFT calculations. The authors define these as “ideal KNLs”. For the specific composition In_xTaS with $x=1/2$, the authors primarily examine the spin degeneracy of the KNL and its location in momentum space using spin-integrated ARPES. Additionally, the authors employ spin-resolved ARPES to reveal Ising-type spin-valley locking near the H point. These experimental findings are consistently supported by DFT calculations. The authors further investigate the Fermi surfaces of In_xTaS with $x=1$ through quantum oscillation measurements.

The manuscript is written well and provides a thorough exploration of intercalated TaS₂, presenting a wealth of data, including resistivity measurements to study superconductivity, spin-integrated and spin-resolved ARPES to characterize KNLs and spin-valley polarization, as well as theoretical calculations of band structures, spin polarization, and analysis of vortex topology induced by KNLs. As the first study to identify a material exhibiting ideal KNLs and to provide such a comprehensive investigation, the manuscript is suitable for publication in Nature Communications. However, I have a few comments and suggestions for improvement:

Reply: We thank the reviewer for the nice summary of our work and the subsequent comments to help us improve the manuscript for a potential publication at Nature Communications.

(1)The authors state, “the momentum dependent relativistic pseudospin splitting pattern is reminiscent of the recently generalized concept of nonrelativistic altermagnetism”. It should be careful about this statement since the KNL does not break time-reversal symmetry, but altermagnet does.

Reply: We completely agree with the reviewer that KNL does not break time-reversal symmetry, but altermagnetism does. It is important to clarify that. Therefore, we have modified the description in the main text to be “of which the momentum-dependent relativistic pseudospin splitting pattern is reminiscent of the recently generalized concept of nonrelativistic altermagnetism^{38,39} but achieved without time reversal symmetry breaking. Meanwhile, the fact that the directional relativistic splitting is locked with the crystal structure symmetry, unlike the nonrelativistic altermagnetic splitting associated with the ligand-environment-enriched antiferromagnetism, enables experimental investigation without complications of domain alignment.”

The goal of the quoted statement in the “Introduction” is to emphasize that the spin polarization probed in the KNL materials does not suffer from domain averaging issue that is challenging to reliably control in antiferromagnetism-based materials.

(2)The connection between spin-valley polarization and KNLs is not sufficiently clear in the manuscript. While KNLs arise from non-symmorphic symmetries, spin-valley polarization typically results from spin-orbit coupling. The authors should provide a more detailed explanation of how KNLs contribute to or interact with spin-valley polarization. Specifically, does the existence of KNLs play a direct role in generating spin-valley polarization, or are these phenomena independently driven by different mechanisms? Further elaboration on this relationship would strengthen the manuscript.

Reply: We thank the reviewer very much for this comment. We acknowledge that the previous version of manuscript did not clarify the underlying relation among KNLs, spin-valley polarization, and the Ising-type spin-orbit coupling (SOC). We clarify that Kramers nodal lines are not the driving mechanism of spin-valley polarization. Inversion symmetry breaking is the essential ingredient for the three phenomena of interest. Theoretically, the KNL is defined for noncentrosymmetric achiral crystals, requiring broken inversion symmetry and the presence of mirror or roto-inversion symmetries, as detailed in Xie, Y. M. et al. Nat Commun. **12**, 3064 (2021). This does not rely on nonsymmorphic crystalline symmetries. Regarding spin-valley polarization, it was pointed out in Xiao, D. et al. Phys. Rev. Lett. **108**, 196802 (2012) that “inversion symmetry breaking together with spin-orbit coupling leads to coupled spin and valley physics in monolayer of MoS₂ and other group-VI dichalcogenides”. Therefore, in our setting, the indium intercalation serves the essential role of breaking the inversion symmetry in In_xTaS₂ and heavy elements such as Ta and In provide the strong strength of SOC. The resultant crystal structure exhibits a horizontal M_z mirror and the vertical mirrors of the Γ MLA planes related to each other by the C₃ rotational symmetry. Under time reversal symmetry, these symmetry generators protect the Kramers nodal lines, for example between the two valleys along $\Gamma - M$, concentrating the Berry curvature and enforcing vanishing spin polarization. On the two sides of the Kramers nodal lines, time reversal symmetry requires the spin-valley polarization to switch the sign. In addition, due to the lack of Γ KHA mirror in the achiral little group, electron spins are pinned to be out-of-plane, termed as the Ising SOC, as explained in He, W.-Y et al. Commun. Phys. **1**, 40 (2018) and Zhou, B. T. et al. Phys. Rev. B. **93**, 180501(R) (2016) by allowing a

$\beta_{so}\sigma_z$ term, unlike the Rashba-type SOC where electron spins are pinned within the 2D plane.

Therefore, we have inserted such statement on the second paragraph of the section “Ideal Kramers nodal line metal $\text{In}_{1/2}\text{TaS}_2$ with spin-valley polarization” before presenting the spin-ARPES data: “In the $\text{In}_{1/2}\text{TaS}_2$ family, the spin-valley polarization arises from the inversion symmetry breaking³⁷ which is satisfied as a prerequisite of the KNL little group symmetries. Meanwhile, the mirror symmetries such as the magenta ones denoted in Fig. 1(b) in the noncentrosymmetric achiral little group generate KNLs concentrating Berry curvature and forcing spin degeneracy robust against SOC. Further, heavy elements such as Ta provide strong strength of the SOC, exhibiting an Ising-type splitting at valleys such as K and K' mainly due to the broken $\Gamma - K - H - A$ mirror⁵⁷.”

Furthermore, we modified the statement in the section of “Discussion and Conclusions” to be: “The spin-orbital texture in $\text{In}_{1/2}\text{TaS}_2$ directly observed via spin-resolved ARPES offers a natural explanation for the spin-valley polarization engendered by the underlying broken inversion symmetry of the KNL little group, with a large SOC spin splitting up to around 250 meV.”

(3)The experiment investigation of KNL and spin-valley is mainly focused on In_xTaS_2 with $x=1/2$, while the quantum oscillation measurement is conducted on In_xTaS_2 with $x=1$. There seems exist a discrepancy. To me, the role of KNLs in quantum oscillations, and potentially in magnetic breakdown, is an intriguing aspect that warrants further exploration. Given that In_xTaS_2 with $x=1/2$ exhibits the best (quasi-)two-dimensional band structure among the studied samples, it is surprising that the authors did not investigate quantum oscillations in this material. Such measurements could provide valuable insights into the Fermi surface topology and the influence of KNLs on electronic properties.

Reply: We thank the reviewer’s suggestion to investigate the quantum oscillation and potential magnetic breakdown in In_xTaS_2 ($x=1/2$). We agree that these are indeed very interesting to explore.

We performed magnetoresistance measurements with a magnetic field along the c axis. Despite the large and non-saturating magnetoresistance, the quantum oscillations are absent up to 14 T. Due to the nature of the random indium occupancy in the lattice, the In_xTaS_2 crystals with $x=1/2$ have more disorder than those with

$x=1$. This can be inferred from the resistivity measurements in the main text Fig. 1(f), where the residual resistivity ratio $RRR = 55$ for $x=1$ and 4.3 for $x=1/2$, as well as the smaller magnetoresistance shown below in Fig. R1(a) for $\text{In}_{1/2}\text{TaS}_2$. The quantum oscillations might be too small to be observed in these crystals with disorder and lower scattering lifetime.

As shown in Fig. R1(b), we also performed resistivity measurements on $\text{In}_{1/2}\text{TaS}_2$ up to 43.8 T at the National High Magnetic Field Laboratory and did not observe quantum oscillations at the base temperature of 0.35 K.

Figure R1. Magnetoresistance (a) and resistivity (b) measurements of $\text{In}_{1/2}\text{TaS}_2$ under a magnetic field along the out-of-plane c axis.

(4) Given that the authors successfully fit the ARPES data using DFT calculations, it would be highly beneficial to construct a tight-binding model that captures the essential features of the bands near the Fermi level, including the Kramers nodal lines (KNLs) and spin-valley polarization. For example, the Figure for KNL is done by toy model in Fig. 1d. Such a model would serve as a valuable starting point for future studies, particularly in exploring the interplay between superconductivity and the topological features of the material.

Reply: We thank the reviewer for the comment on constructing a low-energy tight-binding model. The band around Fermi surface of bulk $\text{In}_{1/2}\text{TaS}_2$ is dominated by d_{z^2} , d_{xy} and $d_{x^2-y^2}$ orbitals from Ta atoms. To capture the essential features of the bands at the Fermi energy, the Kramers nodal lines and the spin-valley polarization as pointed out by the reviewer, we can construct an effective two-band model which

also captures the symmetry of the lattice (the D_{3h} point group symmetry). Here, we use a rotationally symmetric basis to construct an effective two-band model. Based on symmetry analysis, the effective Hamiltonian up to the third-nearest-neighbor is described by

$$H = H_0 + H_{SOC}$$

Here, H_0 is the Hamiltonian with hopping up to the third order:

$$\begin{aligned} H_0 = & \epsilon_0 + 2t_{xy,1}[2 \cos \alpha \cos \beta + \cos(2\alpha)] \\ & + 2t_{xy,2}[2 \cos(3\alpha) \cos(\beta + \phi) + \cos(2\beta - \phi)] \\ & + 2t_{xy,3}[2 \cos(2\alpha) \cos(2\beta) + \cos(4\alpha)] \\ & + \{2t_{z,1} + 4t_{z,2}[2 \cos \alpha \cos \beta + \cos(2\alpha)] \\ & + 4t_{z,3}[2 \cos(3\alpha) \cos(\beta + \phi_z) + \cos(2\beta - \phi_z)]\} \cos(\gamma) \end{aligned}$$

H_{SOC} represents the Ising type SOC:

$$H_{SOC} = -2\lambda[\sin 2\alpha - 2 \sin(\alpha) \cos(\beta)]\sigma_z$$

with the definition $(\alpha, \beta, \gamma) = \left(\frac{k_x a}{2}, \frac{\sqrt{3}k_y a}{2}, k_z c\right)$, where a and c are the lattice constants in the x-y plane and in the z-direction respectively. The specific parameters in this third-nearest-neighbor tight binding model can be fitted from the Bloch spectral function calculated by the Green's function-based density functional theory and are summarized in the following table:

ϵ_0 (eV)	$t_{xy,1}$ (eV)	$t_{xy,2}$ (eV)	$t_{xy,3}$ (eV)	$t_{z,1}$ (eV)
-0.2181	0.0553	0.1165	-0.0271	-0.0326
$t_{z,2}$ (eV)	$t_{z,3}$ (eV)	ϕ	ϕ_z	λ (eV)
0.0205	0.0122	-0.0342	0.3327	-0.027

Based on this tight binding model, one can plot the Fermi contours and Kramers nodal lines. In the Fig. R2(a), we plot the Fermi contours at $k_z = 0$ plane, one can observe the spin polarization in two valleys as observed in the experiment. In Fig. R2(b), we plot the Kramers nodal lines on the $k_z = 0$ plane, which are protected by the mirror symmetry M_z and the time-reversal symmetry. Three Kramers nodal lines connected by three-fold rotational symmetry are shown in white lines in Fig. R2(b).

Figure R2. (a) The Fermi contours that show the Ising type spin polarization at $k_z = 0$ plane. (b) The energy difference of two selected Ising SOC-split bands $|E_\uparrow(k) - E_\downarrow(k)|$ in the $k_z = 0$ plane (in units of eV). The white lines are the Kramers nodal lines at which the two spin bands are degenerate along those lines.

As envisioned by the reviewer, one can use this minimal two-band model for further studying the superconducting and topological properties of the material. More complicated models (such as three band models starting from the d-orbitals of the Ta atoms) can also be constructed, but this minimal two-band model is sufficient to capture the properties of the materials near the Fermi energy.

A list of major changes made to the main text and the supplementary

Position of changes	Brief description
Main text Figs. 1(a, b, c, d, e)	Denote important symmetries for the KNLs, modify KNL annotation and contrast, replace InTaS ₂ 3D Fermi surfaces with the In _{1/2} TaS ₂ ones
Main text Fig. 4	Keep only the ARPES part of InTaS ₂
Main text Fig. 5	Move the 3D Fermi surface of InTaS ₂ (original Fig. 1(e)) and the quantum oscillation results to Fig. 5
Supplementary Note 3 and Supplementary Fig. 3	Spatial homogeneity scan and elemental analysis of the ideal Kramers nodal line semimetal candidate In _{1/2} TaS ₂ , using SEM-EDX

Supplementary Note 4, Supplementary Fig. 4, and Supplementary Fig. 5	Photon-energy-dependent ARPES measurements of $\text{In}_{1/2}\text{TaS}_2$ and InTaS_2 , supported by first-principles calculations
Supplementary Note 5, Supplementary Fig. 6, and Supplementary Fig. 7	In and S termination-dependent ARPES and first-principles studies on InTaS_2 and $\text{In}_{1/2}\text{TaS}_2$
Main text section “Introduction” the 3 rd paragraph	Emphasize that the spin splitting in the Kramers nodal line metal does not break time reversal symmetry, unlike altermagnetism
Main text section “Ideal Kramers nodal line metal $\text{In}_{1/2}\text{TaS}_2$ with spin-valley polarization” the 2 nd paragraph	Explain how Kramers nodal lines, spin-valley polarization, and the strong Ising-type spin-orbit coupling arise and are associated with each other from the underlying noncentrosymmetric achiral little group symmetries and the chemical environment of atoms
Main text section “Introduction” the 3 rd paragraph, and Fig. 1(b) left	Emphasize only $x=1/2$ and 1 for In_xTaS_2 , while pointing interested readers to the supplementary for $\text{In}_{2/3}\text{TaS}_2$ and PbTaSe_2

KNL: Kramers nodal line.

ARPES: angle-resolved photoemission spectroscopy.

SEM-EDX: scanning electron microscopy with energy dispersive x-ray spectroscopy.

Reviewer 1

The authors have addressed all the comments well and made appropriate revisions accordingly. I recommend this manuscript for publication in Nature Communications in its current form.

Reply: We thank the reviewer for recommending the publication of this manuscript at Nature Communications in its current form.

Reviewer 3

All the comments are addressed, so I suggest the current form published in Nat Comm.

Reply: We thank the reviewer for suggesting publishing the current form of the manuscript in Nature Communications.